# Potential erosion capacity of gravity currents created by changing initial conditions

Zordan Jessica[1], Schleiss Anton J.[1], and Franca Mário J.[1,2]

[1]Laboratory of hydraulic constructions (LCH), School of Architecture, Civil and Environmental Engineering, École Polytechnique Fédérale de Lausanne (EPFL), Lausanne, Switzerland
[2]Water Science and Engineering department, IHE Delft Institute for Water Education and Department of Hydraulic Engineering, Delft University of Technology, Delft, The Netherlands

*Correspondence to:* Zordan Jessica (jessica.zordan@epfl.ch)

**Abstract.**

We investigate to which extent initial conditions (in terms of buoyancy and geometry) of saline gravity currents flowing over a horizontal bottom influence their run-out and entrainment capacity. In particular, to which extent the effect of the introduction of an inclined channel reach, just upstream from the lock gate, influences the hydrodynamics of gravity currents and consequently its potential erosion capacity is still an open question. The investigation herein presented focus on the unknown effects of an inclined lock on the geometry of the current, on the streamwise velocity, on bed shear stress and on the mechanisms of entrainment and mass exchange. Gravity currents were reproduced in laboratory, through the lock-exchange technique, and systematic tests were performed with different initial densities, combined with five initial volumes of release on horizontal and sloped locks. The inclination of the upstream reach of the channel (the lock) was varied from 0% to 16% while the lock-length was reduced up to 1/4 of the initial reference case. We observed that the shape of the current is modified due to the enhanced entrainment of ambient water, being the body the region of the current where this most happens. A counter-intuitive relation between slope and mean streamwise velocity was found, supporting previous findings which hypothesized that gravity currents flowing down small slopes experience an initial acceleration followed by a deceleration. For the steepest slope tested, two opposite mechanisms of mas exchange are identified and discussed, i.e. the current entrainment of water from the upper surface due to the enhanced friction at the interface and the head feeding by a rear fed current. The bed shear stress and the corresponding potential erosion capacity are discussed giving insights into the geomorphological implications of natural gravity currents caused in different topographies settings

## 1 Introduction

Gravity currents are common phenomena which may occur spontaneously in nature or triggered by human activities. These flows are created by differences in hydrostatic pressures at the surface of contact of two fluids which have different densities. Examples of gravity currents generated in the atmosphere are katabatic winds which are created by temperature inhomogeneities that originate the density gradient. Avalanches of airborne snow, plumes of pyroclasts from volcanic eruptions are atmospheric flows where suspended particles play a major role in producing the density gradient. If suspended sediment pro-

duces the extra density, gravity currents take the name of turbidity currents. Turbidity currents have received major attentions since the sedimentation they induce, due to the high amount of sediments they transport, have important economic costs consequent to the loss of volume for water storage (Palmieri et al. (2001), Schleiss et al. (2016)). Among gravity currents caused by human actions, the release of pollutants into rivers, oil spillage in the ocean and the desalination plants outflows are of primary importance due to their negative environmental impacts.

Gravity currents have been subjects of research over the last decades. Simpson (1997), Kneller and Buckee (2000), Huppert (2006) and Ungarish (2009) present a comprehensive review of the early work on natural and experimentally reproduced gravity currents. Recently Azpiroz-Zabala et al. (2017) provided a new model for the gravity current structure. They argued that real world turbidity currents in submarine canyons are characterized by a so-called "frontal-cell" which is highly erosive and therefore able to be self-sustaining and to outrun the slower moving body of the flow, creating a stretched current. Nevertheless, authors working on small scale experimentally reproduced gravity currents agree on describing the shape of the gravity current as composed by an arising highly turbulent front, called head, followed by, in some cases, a body and a tail. The difference between the two concepts mainly comes from the observation time-frame which is of the order of hours for the experimentally reproduced gravity currents while it's of days for the observations that Azpiroz-Zabala et al. (2017) made in Congo Canyon.

Within the body, that can reach a quasi-steady state, a vertical structure can be distinguished. A gravity current presents two main interfaces where exchanges concur: at the bottom, generally a solid boundary, and at the top, at the interface with the ambient fluid. These are active boundaries where mass and momentum exchanges are promoted (Ancey, 2012). Ambient fluid is entrained due to shear and buoyancy instabilities at the upper interface (Cantero et al., 2008) resulting in the dilution of the underlying current and modification of the density profile which characterizes a gravity current under stable density stratification (Turner, 1973). If the gravity current travels above an erodible bed, entrainment of material from the bottom can take place, which is conveyed with the current and redeposited sometimes at large distances from their original position (Zordan et al., 2018a). High shear stress associated with intense ejection and burst events influence erosion and bed load transport (Niño and Garcia (1996), Cantero et al. (2008), Zordan et al. (2018a)). For example, in the shallow shelf region of the lake it is frequently observed that cold water, relatively denser than that in open waters, starts to descend down the slope as a cold gravity current (Fer et al., 2002). The plume is able to transport suspended sediment together with their dissolved components, oxygen, and pollutants into deeper water. A proper parametrization of both upper layer and bottom entrainment is still an open research field which needs to be addressed. Indeed, small variations in the entrainment highly influence the flow dynamics (Traer et al., 2012).

Due to the instabilities at the interface with the ambient fluid, the current entrains the lighter fluid and therefore it dilutes. Lock-volume and lock-slope are initial trigger conditions of the experimentally reproduced gravity currents and the main objective of the paper is to understand their influence into the transport capacity of the flows. We show how shear stress at the boundaries is dependent to the set-up under which a gravity current forms, i.e. its initial and boundary conditions. Different initial conditions, representing configurations which can possibly be found in nature, are tested by varying the initial volume of denser fluid, and the lock geometry.

Gravity currents are here reproduced in laboratory by the lock-exchange technique. Three initial densities are tested in combination with five lock-lengths on horizontal bottom and with four inclinations of the upstream channel reach. The bottom of the channel was designed in order to have a variable slope angle of the lock and a following flat surface. We were in search for a threshold at which an inversion of the leading forces of these currents would occur, which are gravitational forces and friction at the upper interface with the ambient fluid. Previous studies mainly focused separately on either low slopes or large slopes, missing the analysis of the transition which is here tested thanks to a specific experimental set-up which allows a wider range of configurations. Finally, we use a parameter previously defined in Zordan et al. (2018a) for the evaluation of the bottom erosion capacity, as a surrogate to evaluate the influence of each different trigger condition on the erosion capacity of the currents.

Britter and Linden (1980) reproduced gravity currents down a slope with no breaks and he found a critical angle, which is typically less than a degree, over which buoyancy force is large enough to counter-act the bottom friction producing a steady flow. At larger slopes, two mechanisms affect the evolution of the current: the current entrains water from the upper surface due to the enhanced shear stress and the head is fed by the rear steady current. Mulder and Alexander (2001) studied slope-break deposits created by turbidity currents. They said that the amount of mixing between flow and ambient fluid is influenced by slope changes which furthermore cause significant changes in turbidite thickness. In the present study the effect of a change in slope is analysed by testing a range of lock-slopes below 16% ($0° \leq \alpha < 9°$). It is expected that the two mechanisms mentioned by Britter and Linden (1980) take place in the lock for the depletion current here formed, due to the incremental gravitational forces, so the transition from a friction governed flow to a flow in which gravitational forces become more and more important happens. The erosion potential of a gravity current formed under such varying initial conditions is then discussed.

The present paper is structured as such: first the experimental set-up and the process which allows for the noise reduction of the velocity measurements are described. Then, the results are presented: a method for the identification of the shape of the current is described and, by means of the mean streamwise velocity field, both bottom and interface shear stresses are computed. The variation on the shape caused by changing initial conditions (i.e. with different initial buoyancies, various lock-lengths and lock-bottom inclinations) are therefore discussed. The potential water entrainment and bottom erosion capacity are estimated on the base of the computed shear stresses evolutions. Finally, an overview of the main findings is presented in the conclusions.

## 2 Methodology

### 2.1 Experimental set-up

The tests are performed in a channel with a rectangular section, 7.5 m long and 0.275 m wide. The gravity currents are reproduced through the lock-exchange technique by sudden release of a gate which divides the flume in two parts where the fluids of different densities are at rest. Three buoyancy differences are tested in combination with five lock-slopes, ranging from $S = 0\%$ (horizontal bed) to $S = 16\%$ (tests S0 to S4). Figure 1 shows the configurations from horizontal bed (S0) to the steepest slope (S4). By introducing a slope on the channel lock reach, the volume of denser fluid is reduced. This lock

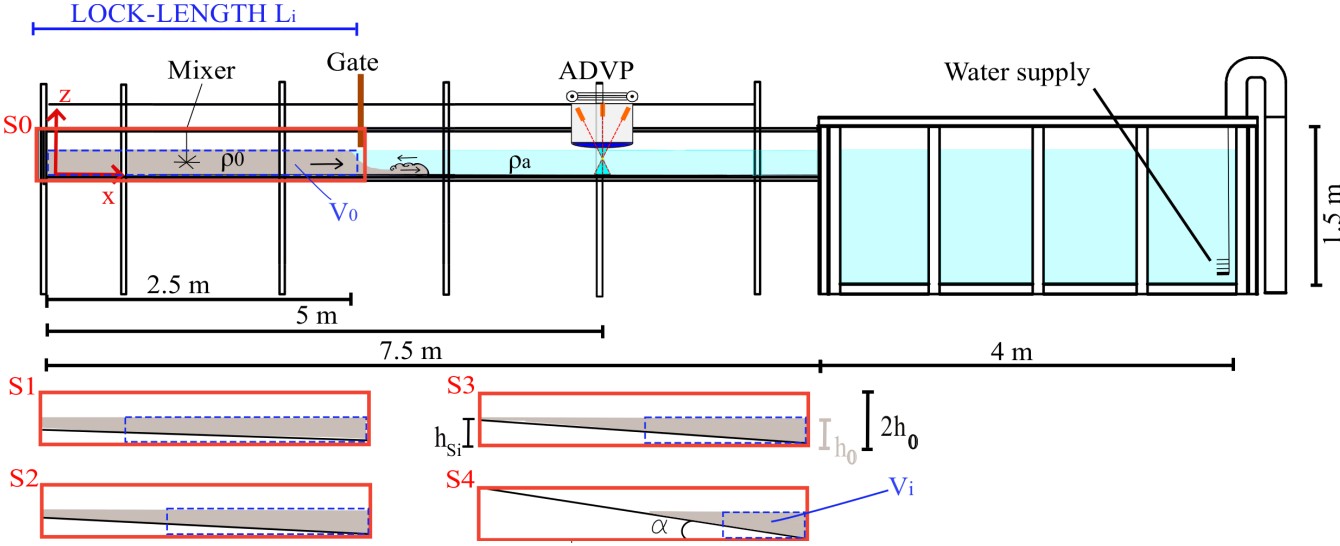

**Figure 1.** Longitudinal view and cross-section of the experimental set-up showing tested slope configurations S0 to S4 of lock volumes $V_i$.

contraction is also tested separately by performing reference tests with the combination of the three initial densities and the five lock-lengths which correspond to the very same volumes in the lock of the tests under inclined. The experimental parameters are reported in Table 1. $R_i.S_i$ refer to gravity currents reproduced by different initial density with the presence of a lock-slope while $R_i.L_i$ indicates the tests with varying initial density and lock-length.

The channel is filled with 0.2 m of ambient water in one side and of salty water, up to the same level, in the lock reach. Once the gate is removed, the saline current forms. At 2.5 m from the gate an Acoustic Doppler Velocity Profiler (ADVP) is placed to measure 3D instantaneous velocities along a vertical. The ADVP (Lemmin and Rolland (1997), Hurther and Lemmin (2001), Franca and Lemmin (2006)) is a non-intrusive sonar instrument that measures the instantaneous velocity profiles using the Doppler effect without the need of calibration and was used with an acquisition frequency of 31.25 Hz. The velocity profiles are collected in time along a fixed vertical. The flume is connected to a final big reservoir that allows the current dissipation and avoids its reflection upstream.

Normalization of the time is made using the scale $t^* = h_b/u_b$, where $h_b$ is a vertical geometric scale, here considered as one third of the total height of the fluid in the experimental tank, $h_0$ ($h_b = h_0/3$ and $h_0 = 0.2$ m) and $u_b = \sqrt{g'h_b}$ is the buoyancy velocity.

## 2.2 Data filtering

By means of the analysis of the power spectra of the raw data collected with the ADVP, noisy frequencies were mainly detected below 8 Hz. The instantaneous measurements were thus low-pass filtered with 8 Hz as cut-off frequency (Zordan et al., 2018a). The 8 Hz cut-off has been chosen because the signal, for frequencies higher than 8 Hz, showed white noise. The time-series of the mean streamwise and vertical velocities ($\overline{u}$ and $\overline{w}$) for the unsteady gravity current, were derived after a

| $S_i$ tests | $\rho_0$ $\left[\frac{kg}{m^3}\right]$ | $g'_0$ $\left[\frac{m^2}{s}\right]$ | $u_0$ $\left[\frac{m}{s}\right]$ | $Re_0$ $[-]$ | $S$ $[\%]$ | $\alpha$ $[°]$ | $V_i/V_0$ |
|---|---|---|---|---|---|---|---|
| R1.S0 | 1028 | 0.29 | 0.24 | 48166 | 0 | 0.00 | 1.000 |
| R1.S1 | 1028 | 0.29 | 0.24 | 48166 | 4 | 2.29 | 0.750 |
| R1.S2 | 1028 | 0.29 | 0.24 | 48166 | 6 | 3.43 | 0.625 |
| R1.S3 | 1028 | 0.29 | 0.24 | 48166 | 8 | 4.57 | 0.500 |
| R1.S4 | 1028 | 0.29 | 0.24 | 48166 | 16 | 9.09 | 0.250 |
| R2.S0 | 1038 | 0.39 | 0.28 | 55857 | 0 | 0.00 | 1.000 |
| R2.S1 | 1038 | 0.39 | 0.28 | 55857 | 4 | 2.29 | 0.750 |
| R2.S2 | 1038 | 0.39 | 0.28 | 55857 | 6 | 3.43 | 0.625 |
| R2.S3 | 1038 | 0.39 | 0.28 | 55857 | 8 | 4.57 | 0.500 |
| R2.S4 | 1038 | 0.39 | 0.28 | 55857 | 16 | 9.09 | 0.250 |
| R3.S0 | 1048 | 0.49 | 0.31 | 62610 | 0 | 0.00 | 1.000 |
| R3.S1 | 1048 | 0.49 | 0.31 | 62610 | 4 | 2.29 | 0.750 |
| R3.S2 | 1048 | 0.49 | 0.31 | 62610 | 6 | 3.43 | 0.625 |
| R3.S3 | 1048 | 0.49 | 0.31 | 62610 | 8 | 4.57 | 0.500 |
| R3.S4 | 1048 | 0.49 | 0.31 | 62610 | 16 | 9.09 | 0.250 |

| $L_i$ tests | $\rho_0$ $\left[\frac{kg}{m^3}\right]$ | $g'_0$ $\left[\frac{m^2}{s}\right]$ | $u_0$ $\left[\frac{m}{s}\right]$ | $Re_0$ $[-]$ | $L$ $[m]$ | $V_i/V_0$ $[-]$ |
|---|---|---|---|---|---|---|
| R1.L0 | 1028 | 0.29 | 0.24 | 48166 | 2.500 | 1.000 |
| R1.L1 | 1028 | 0.29 | 0.24 | 48166 | 1.875 | 0.750 |
| R1.L2 | 1028 | 0.29 | 0.24 | 48166 | 1.563 | 0.625 |
| R1.L3 | 1028 | 0.29 | 0.24 | 48166 | 1.250 | 0.500 |
| R1.L4 | 1028 | 0.29 | 0.24 | 48166 | 0.625 | 0.250 |
| R2.L0 | 1038 | 0.39 | 0.28 | 55857 | 2.500 | 1.000 |
| R2.L1 | 1038 | 0.39 | 0.28 | 55857 | 1.875 | 0.750 |
| R2.L2 | 1038 | 0.39 | 0.28 | 55857 | 1.563 | 0.625 |
| R2.L3 | 1038 | 0.39 | 0.28 | 55857 | 1.250 | 0.500 |
| R2.L4 | 1038 | 0.39 | 0.28 | 55857 | 0.625 | 0.250 |
| R3.L0 | 1048 | 0.49 | 0.31 | 62610 | 2.500 | 1.000 |
| R3.L1 | 1048 | 0.49 | 0.31 | 62610 | 1.875 | 0.750 |
| R3.L2 | 1048 | 0.49 | 0.31 | 62610 | 1.563 | 0.625 |
| R3.L3 | 1048 | 0.49 | 0.31 | 62610 | 1.250 | 0.500 |
| R3.L4 | 1048 | 0.49 | 0.31 | 62610 | 0.625 | 0.250 |

**Table 1.** Experimental parameters. $\rho_0$ is the initial density of the mixture in the upstream tank (measured with a densimeter), $g'$ is the reduced gravity corresponding to $\rho_0$, $Re_0 = u_0 h_0/\nu_c$ is the Reynolds number based on initial quantities with $u_0 = \sqrt{g'h_0}$ the initial buoyancy velocity, $h_0 = 0.2$ m the total height of the water column and $\nu_c$ the kinematic viscosity of the denser fluid, $\alpha$ is the angle of inclination of the bottom in the lock, $S$ is the lock-slope expressed in percentage ($h_{S_i}/L_0$, with $h_{S_i}$ the height as in Figure 1), $L_i$ is the length of the upstream lock-reach, $V_i/V_0$ the percentage of volume of the upstream lock-reach with respect to the configuration $L_0$.

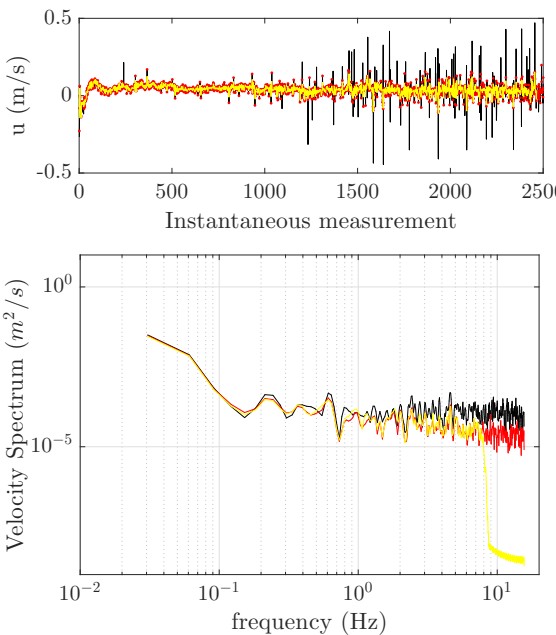

**Figure 2.** Raw velocity data (black) and despiked data (red) obtained with the procedure proposed by Goring and Nikora (2002). Then, through the analysis of the velocity spectra (figure at the bottom), the cut-off frequency of 8 Hz has been identified in order to low pass filter the noisy frequencies (yellow line).

filtering procedure that consisted in the application of a moving average over a time-window which is chosen by the analysis of the power spectra distribution as in Baas et al. (2005). This analysis showed that for a time window of 0.32 s, the harmonics of all the meaningful frequencies were still recognisable while, increasing the time window, the harmonics of progressively smaller frequency gradually lose power and they become impossible to distinguish (Baas et al., 2005). Thus, this window length was chosen for the moving average defining $\overline{u}$ and $\overline{w}$. The turbulent fluctuation time series ($u'$ and $w'$) is then calculated using the Reynolds decomposition:

$$u = \overline{u} + u' \tag{1}$$

where $u$ is the instantaneous velocity. The cleaning procedure with the velocity signals and corresponding spectra is shown in Figure 2.

## 3 Results

### 3.1 The shape of the current

A criterion to identify the two main regions of a gravity current, the head and the body, is here established. Inspired by Nogueira et al. (2014) who considered the product of the depth averaged streamwise velocity with the depth averaged density of the current, we decide to adopt a similar procedure but taking into account the velocity field and the shape as relevant characteristics for distinguishing among head and body. These distinctive features (a characteristic velocity and the contour of the current) have therefore been used in order to define a function which allows to universally identify those regions of the currents. The kinematic function ($H$) is computed as the product between the instantaneous depth averaged streamwise velocity, $u_d(t)$:

$$u_d(t) = \frac{1}{h} \int_0^h u(z,t)dz \tag{2}$$

and the current height, $h(t)$, that is here identified by the position where the streamwise velocity is equal to zero, as in Zordan et al. (2018a). $H$ is thus defined as:

$$H(t) = u_d(t)h(t) \tag{3}$$

By dimensional analysis, the function $H$ corresponds to a flow rate per unit width. The head of the gravity current is characterized by a high specific flow rate which decreases at the rear of the head, a region where fluid is recirculated through vortical movements.

Here $L_h$ identifies the temporal extension of the head and it is identified by the first meaningful local minimum of the function $H$, starting from the front. The conversion from time to length scale may be done by using Taylor frozen hypothesis and considering a reference velocity of the current velocity as advection velocity. This method is coherent for all the experiments performed and the value of the $H$ function is shown in Figure 3.

The body length is instead analysed by using the cumulative sum of the function $H$. This is a region where a quasi-steady regime is established for a certain time length. This implies that $\sum H$ shows a linear increment in time. The limit of the body is therefore defined by analysing the linear evolution that is fitted by a linear regression with least squares method for progressively longer portion of the accumulated summed data. The analysis of the development of $R^2$ value, the coefficient of determination, allows to find the extension of the linear portion which corresponds to the temporal extent of the body region ($L_b$). $R^2$ is defined as the square of the correlation between the response values and the predicted response values. It is

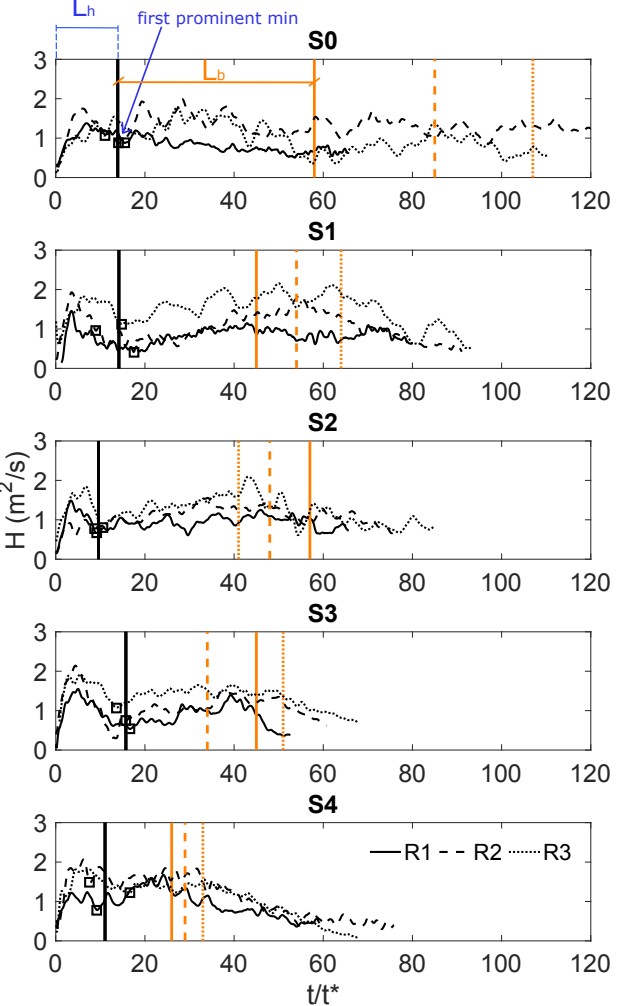

**Figure 3.** Determination of the gravity current head extension from the first prominent minimum of the function H ($L_h$). The extension of the body ($L_b$), as identified by the cumulative sum of the depth-averaged streamwise velocity, is also traced with the red vertical lines.

computed as the ratio of the sum of squares of the regression (SSR) and the total sum of squares (SST) as:

$$R^2 = \frac{SSR}{SST} = \frac{\sum_{i=1}^{n}(y_i - \hat{y}_i)^2}{\sum_{i=1}^{n}(y_i - \bar{y}_i)^2} \tag{4}$$

where $\bar{y}$ is the average of the response $y$, $\hat{y}$ is the regression line and $n$ the number of observations $i$. In Figure 3 the development of the function $H$ is shown for the tests with the lock-slope. The same procedure is adopted for tests $R_i.L_i$ and the results can be find in Zordan et al. (2018b).

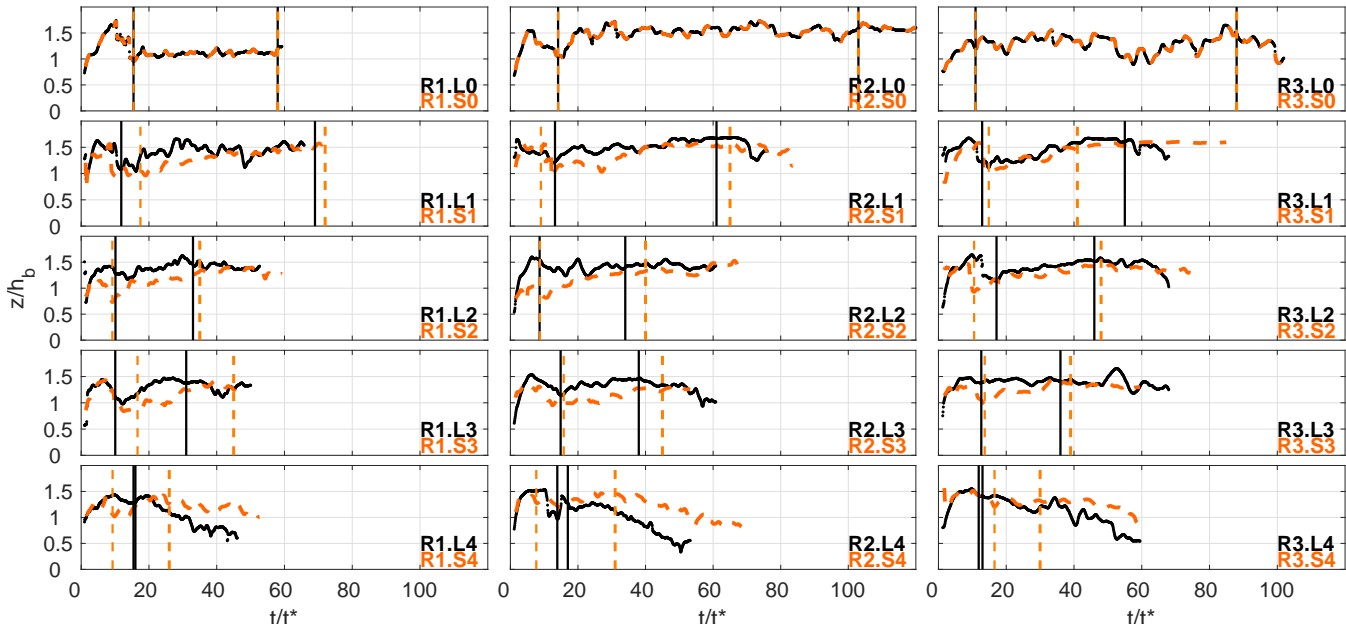

**Figure 4.** Gravity currents contours, as identified by the zero streamwise velocity contour, for tests with the lock-slope and correspondent tests with lock-length variation.

The form of the currents was identified by the zero streamwise velocity contour. In Figure 4 the contours of each test with the lock-slope are compared with the correspondent reference test with lock-length variation. The results are grouped by the initial density on the lock (columns in the figure), and by pairs of tests with the same volume of the lock but for different slopes. The extension of head and body as identified by the previous methods are also reported with the vertical lines. Dashed lines refers to tests $S_i$ while continuous lines correspond to $L_i$ tests.

In Figure 4 we can see that the head of the currents does not show any relevant change. Instead the extension of the body is affected: it reduces with increasing inclination of the upstream channel reach and the same goes for tests produced by reduced lock volume. A dependency on the initial density is noticed and in three out of the total five slopes bring to the formation of longer body with greater initial buoyancy. This can be verified in Figure 3 where extensions of the bodies are plotted with the vertical orange lines. R1, R2 and R3 produces progressively longer bodies for tests S0, S1 and S4.

The largest deviation between the two contours of corresponding tests $L_i$ - $S_i$ is noticed for the last configuration, with the $R_i$.L4 tests showing a shorter body and a more defined tail while for the correspondent tests with the inclined lock the body is more extended.

### 3.2 Mean velocity field

In Figure 5 the mean streamwise velocity field on the background and velocity vectors of the components $(\overline{u}, \overline{w})$ are shown for
all the tests performed. The heads of the currents are indicated by the vertical dashed lines and the zero streamwise velocity

contours are marked by the black lines. We can notice that the structures of the currents are quite similar in all configurations. An arising head is followed by a zone of high mixing, characterized by the presence of billows (due to Kelvin-Helmholtz type of instabilities (Simpson, 1972)) that are due to shear at the rear part of the elevated head. Body and tail are not always well defined regions, mainly for the class of tests down an inclined, and therefore the contour is not drawn. Moreover, tests $S_i$ show lower streamwise velocities within head and body with respect to correspondent $L_i$ tests.

By comparing tests $S_i$ with the correspondent $L_i$ tests, which have the same lock-volume but are performed without upstream slope, it is noticed that mean streamwise velocity is slightly higher for tests on horizontal bed. This can appear to some extent contradictory but that behaviour has already been mentioned in literature, in the study of Beghin et al. (1981), who was one of the first to investigate the role of the slope on the physics of a gravity current. He showed that tests which flows on small slopes, for tests where the entire channel was inclined, (typically less than 5°) experience a first acceleration phase followed by a deceleration phase. This is because of the fact that, although the gravitational force increases as the lock-slope becomes more inclined, there is also increased entrainment, both into the head itself and into the flow behind. This produces an extra dilution of the current with a decrease in buoyancy.

### 3.3 Bottom and upper shear stress

The sedimentological impact of a gravity current is the result of the complex hydrodynamic of this flow. Sediment entrainment is a complex mechanism mainly due to the difficulty in defining the fluctuating nature of turbulent flow (Salim et al., 2017). In Zordan et al. (2018a) the transport of sediment within a gravity current is linked to the bed shear stress, that is here considered a "surrogate" measure of it. The form that bed shear stress is affected by the changing initial conditions of the current explains thus how the entrainment capacity of a current is altered. Bed shear stress temporal evolution is calculated by following the procedure in Zordan et al. (2018a) where it was assumed that the mean flow met the conditions necessary for the fitting of the overlapping layer by the logarithmic law of the wall as (Ferreira et al., 2012):

$$\frac{\overline{u}(z)}{u_*} = \frac{1}{k} \ln \frac{z}{z_0} \tag{5}$$

where $\overline{u}(z)$ is the mean velocity, $u_*$ is the friction velocity, which is the velocity scale corresponding to the bed shear stress (Chassaing, 2010), $k$ is the von Kármán constant, $z$ is the vertical coordinate and $z_0$ is the zero-velocity level.

The equation of the logarithmic law of the wall can be rewritten as:

$$u = A ln(z) - B \tag{6}$$

where

$$A = \frac{u_*}{k}, \quad B = \frac{u_*}{k} ln(z_0) \tag{7}$$

Then, by determining the coefficients $A$ and $B$ through a fitting procedure, one obtains an estimation of $u_*$ which is the velocity scale corresponding to the bed shear stress.

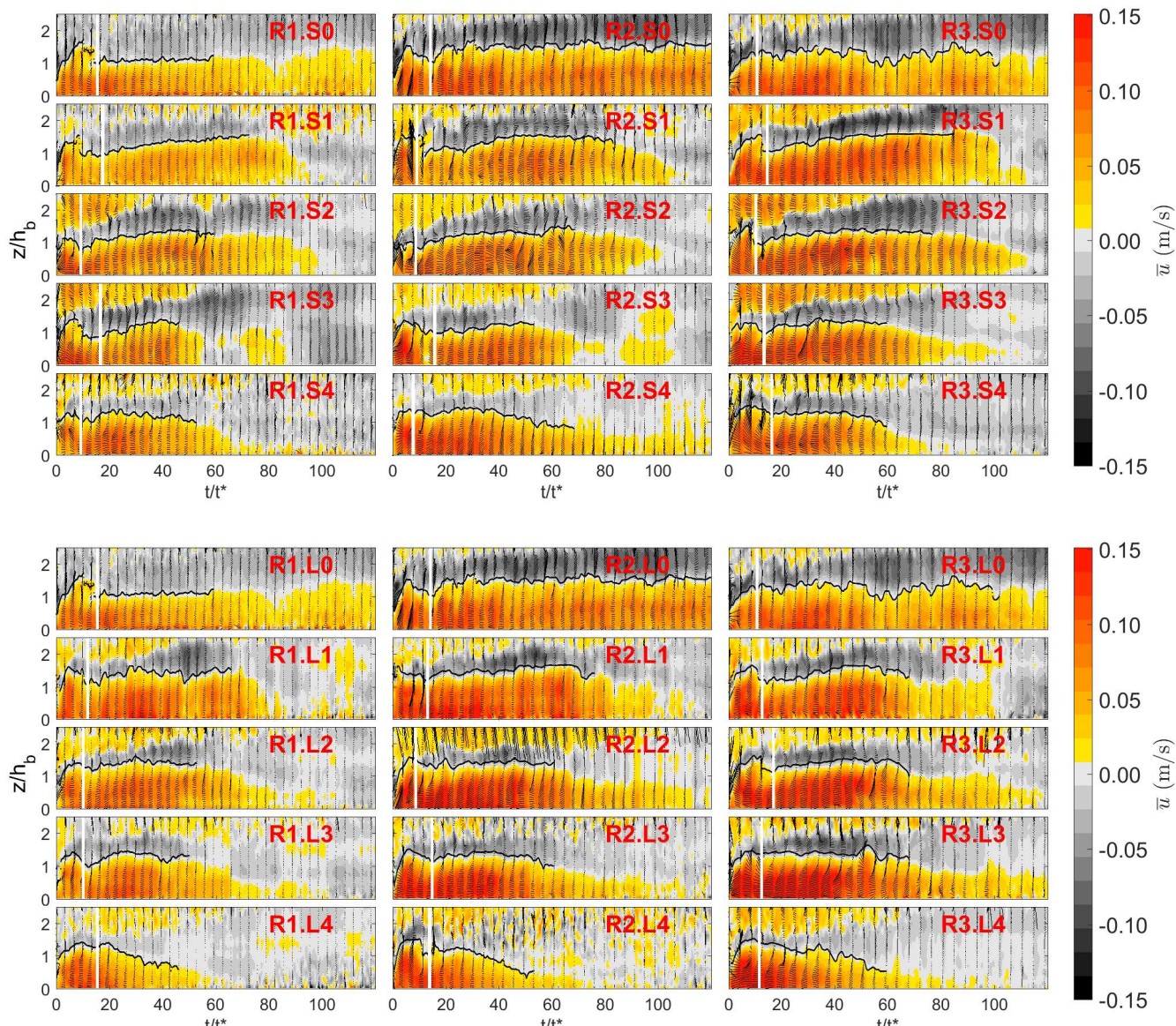

**Figure 5.** Streamwise velocity field on the background and velocity vectors of the components (u,w). The head of the current is delimited by the vertical white line. The contour of the current is indicated in black.

Bed shear stress is afterwards computed by considering a constant initial density that is here equal to the initial density in the lock ($\rho_0$):

$$\tau_b = \rho_0 u_*^2 \tag{8}$$

10 The fitting procedure of the bottom logarithmic layer was determined stepwise: extending a linear least square fitting range (in a semi-logarithmic scale) from the lowest measured point until the point along the vertical of maximum streamwise velocity. Then, within this region, the sublayer which provided the best regression coefficient was chosen and considered for the estimation of $u_*$, corresponding to the extent of the logarithmic layer as it was shown in Zordan et al. (2016).

The flow boundary is assumed to be smooth by verifying that the shear Reynolds number (or skin roughness, $k_s$, normalized by the viscous layer) is lesser than 5 (Nezu et al., 1994):

$$\frac{k_s u_*}{\nu} \leq 5 \tag{9}$$

The classic value of the von Kármán constant of $k = 0.405$ is adopted. Discussion on the estimation of $k$ can be found in Ferreira (2015).

The bed shear stress time-evolution of gravity currents with lock-slope ($\tau_{b,S}$) are compared to the analogous results for tests with decreasing lock ($\tau_{b,L}$). Therefore the time-averaged bed shear stress has been computed and the ratio $\overline{\tau_{b,L}}/\overline{\tau_{b,S}}$ is

5 shown in Figure 6. Tests performed with a lock-slope show in average lower values of bed shear stress ( $\overline{\tau_{b,L}}/\overline{\tau_{b,S}} \geq 1$ ). By increasing the lock-slope this tendency is less evident and the mean bed shear stress compares for both conditions with varying lock-slope and with different lock-lengths. Moreover, tests performed with the highest density difference seem less affected by changing configuration ($S_i L_i$ with $i = 1, 2, 3, 4$). The detailed time series for this condition are presented in Figure 8, where we can see that from normalized time $t/t^* \simeq 20$, i.e. in the body region, bed shear stress is slightly higher for tests $S_i$ than in

the correspondent $L_i$ tests.

At the upper boundary of the gravity currents, i.e. the interface with the ambient water, studies on turbulent flow near a density interface confirmed that under certain conditions, the turbulent boundary layer theory can be applied as well (Lofquist (1960), Csanady (1978)) and that the "law of the wall" can be used to estimate the shear stress here. By hypothesizing a constant mean value of water viscosity and hydraulically smooth conditions, the estimation of an interface shear stress ($\tau_m$) is made;

qualitatively the estimation that will result is enough for the purpose of the present study. The time evolution of interface shear stress is therefore computed following the same procedure as for the bottom shear stress. In this case the fitting procedure of the logarithmic layer is determined by considering the mixing layer as defined in (Zordan et al., 2018b). This layer is delimited at the top by the zero streamwise velocity contour and at the bottom by the height of the current as defined by the Turner's integral scales (Ellison and Turner, 1959). Within this layer the at-least-three consecutive measurement points along the velocity profile

which were giving the highest $R^2$ were considered for fitting. The time-average of the interface shear stress are compared by means of the ratio $\overline{\tau_{m,L}}/\overline{\tau_{m,S}}$ (Figure 7) showing that in general tests performed with varying lock-lengths present higher values with respect to correspondent tests with lock-slope variation i.e. $\overline{\tau_{m,L}}/\overline{\tau_{m,S}} \geq 1$.

Again the main differences between tests with reduced lock and respective tests with lock-slope are for the fourth configuration which detailed time-series are shown in Figure 9. The steepest slopes present higher values of interface shear stress in

the body region with respect to the correspondent tests with the same initial volume of release but flowing on a horizontal bed.

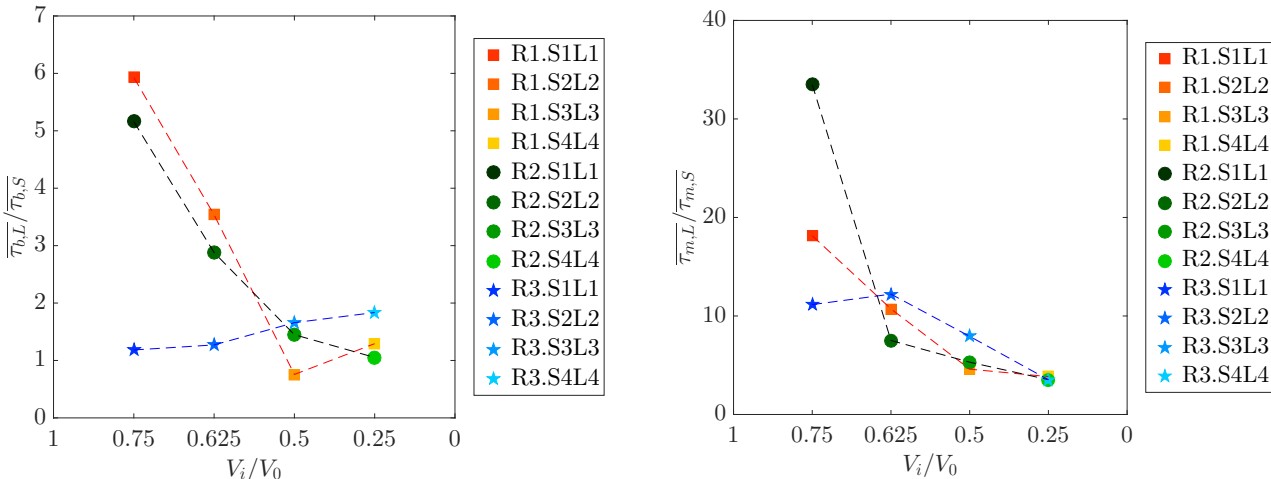

**Figure 6.** Ratio between time-averaged bed shear stress of tests with varying lock-lengths ($\overline{\tau_{b,L}}$) and test with varying lock-slopes ($\overline{\tau_{b,S}}$) versus percentage of volume of the upstream lock-reach $V_i/V_0$. Dashed lines link tests performed with the same initial excess density.

**Figure 7.** Ratio between time-averaged interface shear stress of tests with varying lock-lengths ($\overline{\tau_{m,L}}$) and test with varying lock-slopes ($\overline{\tau_{,S}}$) versus percentage of volume of the upstream lock-reach $V_i/V_0$. Dashed lines link tests performed with the same initial excess density.

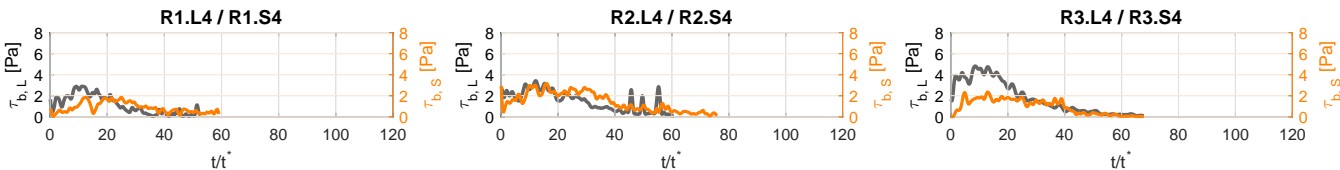

**Figure 8.** Temporal evolution of bed shear stresses calculated by log law fitting for tests with progressively reduced lock-length ($\tau_{b,L}$) and with the lock-slope ($\tau_{b,S}$).

## 4 Discussion

### 4.1 Shape variation of gravity current with the lock-slope

The extensions of the body of correspondent tests performed with the lock-slope or with horizontal bed, and with varying lock-lengths are compared in Figure 10. For lower lock-slopes the body extension is similar to the currents produced with the same lock-volume but with horizontal bottom. However, for lock-slopes at $16\%$, the body region for tests $S_i$ are longer than correspondent tests $L_i$. At this point two mechanisms affect the evolution of the current: the current entrains water from the upper surface due to the enhanced friction at the interface between the denser flow and the counter current progressively advancing upwards the lock, and the head is fed by the rear current. The flow of tests S4 show that the characteristics of the upstream flow in the lock are influencing the flow even when the current reaches the measuring point: (i) an extended body is

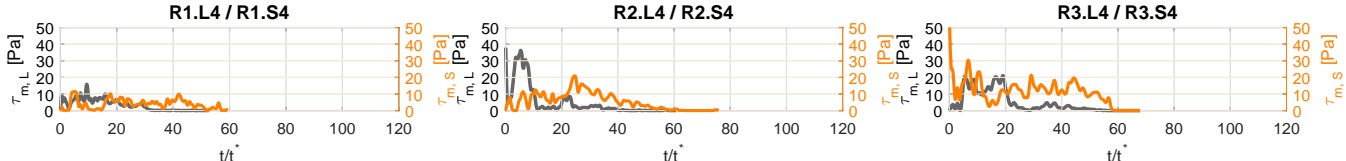

**Figure 9.** Temporal evolution of interfacial shear stresses calculated by log law fitting for tests with progressively reduced lock-length ($\tau_{m,L}$) and with the lock-slope ($\tau_{m,S}$).

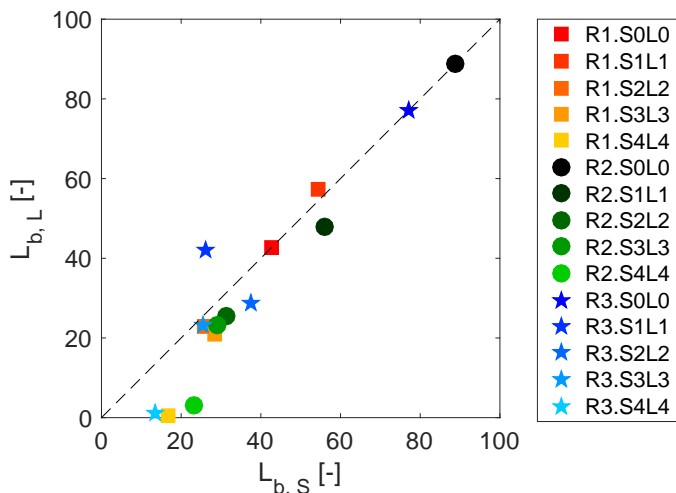

**Figure 10.** Comparison of the length of the body ($L_b$) between tests with progressively reduced lock-length and with lock-slope. The dashed line is the identity line.

the result of water entrainment at the upper surface of the current that creates dilution and expansion of the fluid in the current; (ii) the fluid in the body become faster as a result of the gravitational forces as in Britter and Linden (1980) (Figure 5). Britter and Linden (1980) showed that for currents flowing along a horizontal boundary, the head is the controlling feature. However down a slope, the body becomes more determinant in the gravity current evolution since it is up to 30-40% faster than the head velocity, depending on the slope, being able therefore to move faster fluid into the head. In our study, the lock-slopes 16% show those features and the effect is not only occurring within the inclined lock but is also observed in the downstream flat part of the channel.

## 4.2 Ambient fluid entrainment

Kelvin-Helmholtz instabilities have a major role in provoking water entrainment. They take the form of vortical movements generated due to velocity shear at the interface between the two fluids. Since shear stress is determinant in the process of water entrainment, a new quantity to account for the potential entrainment capacity of the gravity current is here defined on the base

of the computed time evolution of the interfacial shear stress ($\tau_m$). It is computed as the non-dimensional time integral of the shear stress which represents, after dimensional analysis, the work done over a determined duration, per unit surface for a given advection velocity, which can be approximated as the initial buoyancy velocity, $u_0$. This quantity $\Phi_m$ is calculated as:

$$\Phi_m = \int_{T_1}^{T_2} \tau_m(t) dt / t^* \tag{10}$$

where the limits of integration are $T_1 = L_h$ and $T_2 = L_b$, in order to focus on the body, the region that has been found mostly affected by the variation of the initial conditions. The validity of the use of $\Phi_m$ as an indicator of the entrainment capacity is supported by the analysis of its relation with the Richardson number $Ri$ (Zordan et al., 2018b). The relation between water entrainment and bulk Richardson number is well known in literature and numerous empirical fits to the experimental data have been proposed since the early work of Parker et al. (1987) and supported by more recent contributions (Stagnaro and Pittaluga, 2014). Since bulk Richardson number is based on depth averaged quantities, it assumes that properties do not vary significantly along the vertical. The quantity $\Phi_m$, a surrogate for entrainment capacity, relies to the instantaneous measurements of shear stress and therefore account for the unsteady behaviour of the currents. Therefore, it is here proposed to use this quantity as a surrogate for water entrainment capacity since it benefits from the instantaneous measurements of shear stress and it accounts for the unsteady behaviour of the gravity currents. In Figure 11, the potential water entrainments for gravity currents performed on an inclined and correspondent tests with reduced initial volume of release are compared. The tests S4L4 detach from the identity line, thus a greater water entrainment is expected for the case with the inclined bed with respect to the horizontal bottom. The enhanced entrainment which has been verified for gravity currents formed downstream steep slopes is due to the shear at the interface with the ambient water. According to Beghin et al. (1981), gravity currents are experiencing two phases while flowing along the channel. An initial acceleration takes place due to the higher gravitational forces, then the current accelerates inducing an increment of shear stresses at the interface. The entrainment of clear water is therefore intensified and the currents are diluted. At the point where the measurements are taken, the gravity currents are experiencing this second phase.

### 4.3 Bottom erosion capacity

The magnitude of the shear stress at the lower boundary layer determines the sediment transport capacity of saline currents and whether erosion or deposition processes dominate the regime at the bottom boundary (Cossu and Wells, 2012). Therefore, similarly to the interfacial water entrainment capacity, the bottom entrainment capacity, that can also be called the erosion capacity, is here computed on the base of the computed bed shear stress.

This new quantity is defined as:

$$\Phi_b = \int_{T_1}^{T_2} \tau_b(t) dt / t^* \tag{11}$$

where the limit $T_i$ are $T_1 = 0$, $T_2 = L_b$.

In Zordan et al. (2018a) this quantity has been estimated for gravity currents simulated over an erodible bed. A relationship between the eroded volume of sediments provoked by the passage of the gravity current and $\Phi_b$ has been found, therefore

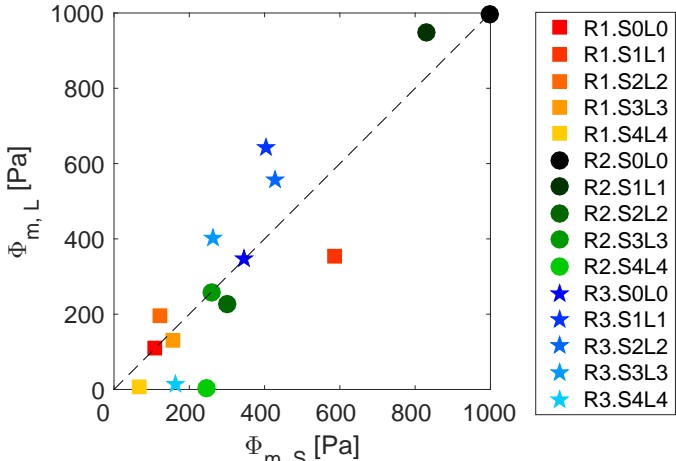

**Figure 11.** Comparison of $\Phi_m$, a surrogate for the entrainment capacity of the mixing region, between tests with lock-slope ($S_i$) and correspondent tests on horizontal bottom ($L_i$). The dashed line is the identity line.

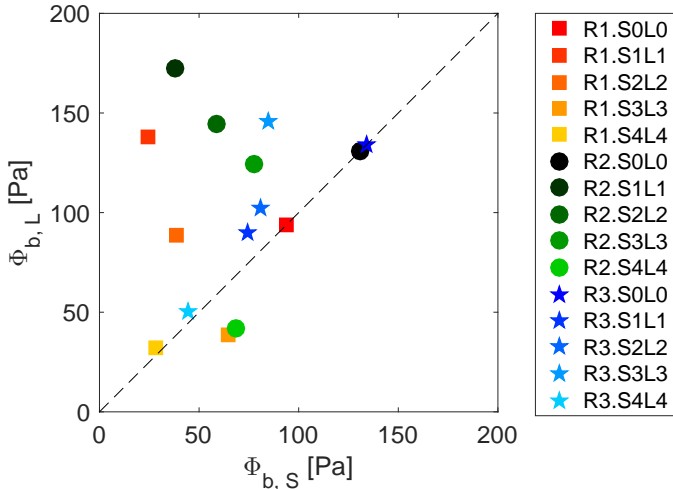

**Figure 12.** Comparison of the bottom erosion capacity $\Phi_b$ between tests with lock-slope ($S_i$) and correspondent tests on horizontal bottom ($L_i$). The dashed line is the identity line.

confirming that $\Phi_b$ is a good estimator of the entrainment capacity of these flows. Although the present experiments are over a fixed bed, this estimator will be used here to evaluate the influence of the lock initial conditions in the entrainment capacity of these flows.

The bottom erosion capacity is compared for gravity currents performed with the lock-slope and correspondent tests on a horizontal bottom. Generally, $L_i$ tests show a higher erosion capacity with respect to their analogous $S_i$. The points in Figure 12 are in fact concentrated above the bisect of the first and third quadrants. The effect of an extra gravitational force occurring in the flow upstream the lock, as described above, is proved not to play a role in enhancing the capacity of the current to perform bottom erosion, in the downstream flat reach of the channel, which is instead reduced. This is probably a consequence of the decrease in streamwise velocity which results from the dilution of the gravity current occurring already in the lock. On the other hand, ambient water entrainment causes the expansion of the body region. Longer bodies keep eroding material longer and the erosion potential attributed to this part is therefore increasing. The potential bottom erosion, i.e. the quantity $\Phi_b$ in Figure 13, shows a tendency to decrease with increasing lock-slope. This is mainly the result of the released volume reduction caused by the presence of the lock-slope, therefore originating shorter current bodies. The role of the body in the total erosion capacity is computed as the ratio $\Phi_{b-body}/\Phi_b$ (Figure 13) whose limits of integration of $\Phi_{b-body}$ are $T_1 = L_h$ and $T_2 = L_b$. The contribution that is ascribed to the body has a similar development as the total erosion capacity. This enforce the hypothesis that the body is determinant in the entrainment capacity of a gravity current. Figure 13 highlights that the importance of the body in the total erosion capacity becomes proportionally higher for tests S4 (the trend lines in Figure 13 deviates more in this configuration). Higher water entrainment was proved in Section 4.2 for this latter case, which was therefore subjected to an expansion of the body region. An influence of the upper surface on the dynamics of the lower bottom boundary is therefore hypothesized. The interaction between the upper layer and the bottom was already pointed out by the numerical investigation of Cantero et al. (2008) and experimental evidences were reported in Zordan et al. (2018b). In this latter the vorticity was analysed showing that residual negative vorticity expands from the upper layer through the bottom with progressively lower intensity.

## 5 Conclusion

In most practical situations gravity currents are flowing on different topographies and most of the time travels along inclined but discontinuous slopes (slope breaks). Moreover they are generally originated by the release of a certain amount of a fluid of various densities. The present study analyses all the previously mentioned changing initial conditions which trigger gravity currents that are commonly observed in nature.

Gravity currents of different initial densities are reproduced experimentally and the effect of incremental gravitational forces (reproduced by using an inclined lock) is analysed. Corresponding tests with a horizontal lock are performed as well in order to have the reference cases with the same reduced volume. The range of lock-slopes tested varies from horizontal bed to $S = 16\%$ (which correspond to an inclination of the lock $\alpha \approx 9°$). The gravitational force is the main driving force which directly depends on slope (Khavasi et al., 2012). Therefore, at the upstream inclined reach which constitutes the lock, there is the action

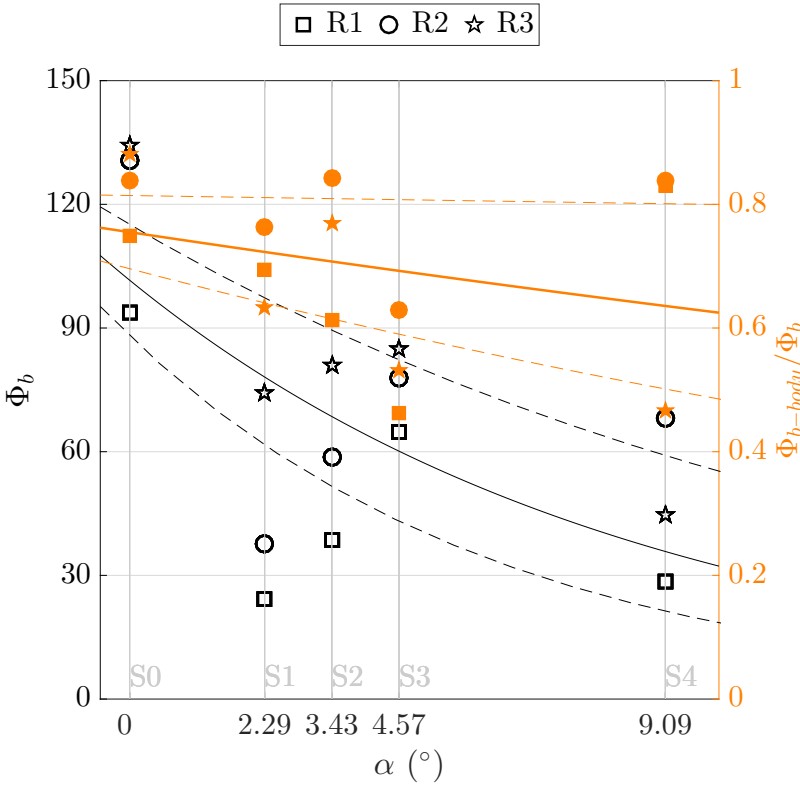

**Figure 13.** Potential erosion capacity for gravity currents (left axis) developed with different lock-inclinations and rate of potential bottom erosion due to the body of the gravity current on the total bottom erosion capacity, $\Phi_{b-body}/\Phi b$, (right axis). The exponential fitting lines are reported in order to give evidence of the general trend, together with the 85% confidence intervals (dashed lines).

of gravitational forces which compete with the entrainment that takes place due to higher shear stress at the upper interface and tend to dilute the current. Thus, if on one hand gravitational acceleration drives for a faster gravity current, on the other hand water entrainment at the upper interface dilute the fluid of the current which is consequently slowed down and it expands due to the incorporation of the ambient fluid. The configurations S4-L4, corresponding to the steepest lock-slope and the shortest lock-length, respectively, exhibit the highest deviations in terms of shape and ambient water entrainment between tests with lock-slopes with respect to correspondent tests on the horizontal bed. S4 tests showed a longer body, owing to entrainment of the ambient fluid. Bottom erosion capacity at the downstream flat reach is reduced by the presence of the extra gravitational forces, most probably due to lower streamwise velocities which are consequence of gravity currents dilution occurring on the way of the gravity current along the channel. The limit case of tests S4, with a lock-slope of $S = 16\%$, is the transient condition as described by previous literature for a continuously sloped channel (Britter and Linden (1980), Beghin et al. (1981), Parker et al. (1987), Maxworthy and Nokes (2007), Maxworthy (2010)), where buoyancy force is large enough to counter-act bottom and upper layer frictions. The limit given by the experimental set up did not allow to go for steeper lock-slopes, cases for which therefore further investigation should be undertaken.

**List of symbols**

Table 2 summarizes the list of symbols used in the paper, their definition and unit of measure.

*Competing interests.* The authors have no conflict of interest to declare.

10  *Acknowledgements.* This research was funded by the European project SEDITRANS funded by Marie Curie Actions, FP7-PEOPLE-2013-ITN-607394 (Multi partner - Initial Training Networks).

| Symbol | Definition | Units |
|---|---|---|
| $A, B$ | coefficients of the log fit | — |
| $g'$ | reduced gravity | $m/s^2$ |
| $h_0$ | total height of the water column | $m$ |
| $h_b$ | vertical geometry scale | $m$ |
| $h_{S_i}$ | vertical elevation of the inclined bottom | $m$ |
| $h(t)$ | current height | $m$ |
| $H(t)$ | kinematic function | $m^2/s$ |
| $k$ | von Kármám constant | — |
| $k_s$ | skin roughness | $mm$ |
| $L_b$ | temporal length of the body | $s$ |
| $L_h$ | temporal length of the head | $s$ |
| $L$ | lock-length | $m$ |
| $L_i$ | test performed with lock-length $i$ | — |
| $n$ | total number of observation | — |
| $R$ | correlation coefficient | — |
| $R_i$ | test performed with initial density $\rho_i$ | — |
| $Re_0$ | Reynolds number based on initial quantities | — |
| $SSR$ | sum of square of the regression | — |
| $SST$ | total sum of square | — |
| $S$ | slope | $\%$ |
| $S_i$ | test performed with slope $i$ | — |
| $t^*$ | normalized time scale | — |
| $u$ | instantaneous streamwise velocity | $m/s$ |
| $u_*$ | friction velocity | $m/s$ |
| $\overline{u}$ | mean streamwise velocity | $m/s$ |
| $u'$ | streamwise turbulent fluctuation | $m/s$ |
| $u_0$ | initial buoyancy velocity | $m/s$ |
| $u_b$ | buoyancy velocity | $m/s$ |
| $u_d$ | depth averaged streamwise velocity | $m/s$ |
| $V_i$ | initial volume of dense fluid released | $m^3$ |
| $w$ | instantaneous vertical velocity | $m/s$ |
| $\overline{w}$ | mean vertical velocity | $m/s$ |
| $w'$ | vertical turbulent fluctuation | $m/s$ |
| $y_i$ | response of the observation $i$ | — |
| $\overline{y}_i$ | average of the response of the observation $i$ | — |
| $\hat{y}_i$ | regression line of the response of the observation $i$ | — |
| $z_0$ | zero velocity level | $m$ |
| | | |
| $\alpha$ | angle of inclination of the bottom | $^\circ$ |
| $\nu$ | kinematic viscosity | $m^2/s$ |
| $\Phi_b$ | bottom entrainment capacity | — |
| $\Phi_m$ | interface entrainment capacity | — |
| $\Phi_{b-body}$ | bottom entrainment capacity due to the body | — |
| $\rho_0$ | gravity current initial density | $kg/m^3$ |
| $\rho_a$ | ambient fluid density | $kg/m^3$ |
| $\tau_b$ | bed shear stress | $Pa$ |
| $\tau_m$ | interface shear stress | $Pa$ |

**Table 2.** List of symbols with definition and unit of measure.

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
