# Peer review of "Potential erosion capacity of gravity currents created by changing initial conditions"

_Earth Surface Dynamics, 2017_

## Referee Comment (RC1) · O. Sequeiros (Referee) · 28 Dec 2017

O. Sequeiros (Referee)

oeseq@yahoo.com

General comments 1) The word "Geomorphic" is very appealing. However it is only used once in the Title and once in the Abstract. It is never used in the text, much less defined. It seems that "geomorphic" is just a fancy word for sediment transport/erosional processes. If you insist in keeping it, then at least make sure you define it and use it in the body text. Don't just place it in the title for attention-catching purposes. 2) I understand that the slope in the channel (as opposed to the slope in the lock) is always zero in these experiments. If this is true, then a lot of the references, e.g. Britter and Linden (1908), could be off the mark or misleading. B&L did experiments with the slope of the channel being the same as in the lock. Your set up is completely different. There is a slope break in your experiments, and you should not try to back your results

up with previous experimental work that have a different set up. This does not invalidate your work or even your analysis, but you need to explain your work in a different way. This pervades many of your interpretations and discussions. 3) Discussion about the shape of the current, Equation 3.Why not just using H=h(t) to tell the body from the head? You might have some reasons, but they are not explained. It comes up as arbitrary the force the process based on H=ud(t)h(t) without further explanation. Figure 3 shows that H can sometimes not be a good parameter to tell head from body, e.g. the top plot corresponding to S0, where the first prominent minimum is not evident at all. 4) Explain better the process of filtering and Figure 2. It is not intuitive how the 8Hz threshold is chosen. In Figure 2, it appears that you are filtering above 8Hz, but the text (P2,L93) hints that you are filtering below it. 5) P5,L36-48. "Mean streamwise velocity is slightly higher for tests on horizontal bed. . ." This is also explained because there are lock-exchange tests, with a finite/fixed volume. In sustained density currents the opposite is true. You need to explain the context of your experiments and explain that lock-exchange flows a subset of something bigger. Otherwise statements like the above can be interpreted to be universal, which is not true. 6) P5,L61. "Under the assumption of flow gradually varied in the longitudinal direction". Do you mean "transient flow" instead of "gradually varied flow"? "Gradually varied" brings to mind something that changes in space, and not necessary in time. In fact a steady non-uniform flow is a gradually varied flow. But your flows are first and most transient, the spatial variation comes as a result of that. 7) P5,L71. "the flow boundary is assumed to be smooth" Why do you have to assume it? Can't you show with the Shields parameter (a function of the shear velocity) and the particle size show that you are in a hydraulically smooth region without assuming anything? See for example Chapter 2 of the Sedimentation Engineering (Manual 110) edited by Garcia (2008) 8) P5,L75-86. You mention that the fitting is done "from the lowest measured point until the maximum velocity vertical position"; this is questionable because the log profile most probably will not be valid near the position of the maximum velocity. I recommend that you show in an Appendix an example on how the fit is actually done. I have done this before and I am aware

that the fitting of the log profile to derive the shear velocity is extremely sensitive to z0 and to which points in the measured data you chose for the fit. 9) P5,L90. "Tests performed on an inclined bottom show..." Here the reader can be misled to believe that the bottom was inclined in the entire channel. See point 2 above. 10) P5,L95ss. Chikita et al. (1991) have an alternative method to account for the effects of the interface shear stress. It could be useful to compare with the approach you have followed. 11) P6,L36. "...and the head is fed by the rear steady current" But your flow is not steady at all, as in a sustained density current. Is inherently transient due to the small volumes released from the locks... 12) P9,L24-25. "The potential bottom erosion, i.e. the quantity Psib in Figure 11, show a tendency to decrease with increasing slope". This is not intuitive at all... Back to my point 2, you are thinking in terms of "slope", but what matters is "volume". The slope of the lock is misleading in order to understand this behavior. You are describing phenomena that happen in the zero slope-channel, downstream from the lock. I would rather venture that because tests with steeper slope have smaller volume, their flows downstream the channel (at least where you measure shear stress) have smaller power/intensity and eventually lower shear stress. It is a about volume, not the slope of the lock. Please reconsider. 13) P10,L32-34. "The configurations S4-L4, corresponding to the steepest lock-slope and the shortest lock-length, respectively, exhibit the highest deviations between tests with lock-slopes with respect to correspondent tests on the horizontal bed" Same as last point. This makes sense because S4 tests should be seen as small volume tests, rather than anything else. Downstream the channel, away from the lock, the slope of the lock area is arguably a second order effect. 14) P10, L27-31. "Bottom erosion capacity generally results reduced by the presence of the extra gravitational forces most probably due to lower streamwise velocities which followed gravity currents dilution." Reword this sentence. Difficult to follow in its current form. 15) P10, L38-40. "The present study analyses changing initial conditions which trigger gravity currents that are commonly observed in nature." What conditions are commonly observed in nature?

Minor/format comments 1) Abstract: "The shape of the current is altered due to the

**ESurfD**
enhanced entrainment of ambient water and mainly the body of the current results affected." Reword this sentence. 2) Abstract, last sentence: What is "a base experience"? Reword. 3) Page , Line 27-28: Not sure what is the meaning of "in same cases" in the middle of the sentence. Reword. 4) P2, L3. It is "Niño and García", not "Ninto and Garcia". 5) Table 1. Caption. $\Delta T$, $T0$, $Ta$ do not appear in the Table. 6) P4,L9. Define R in R^2. 7) P6,L36. What do you mean by "faster material"? Reword. 8) P7,L3 and elsewhere. T1, T2 are time parameters, while Lh, Lb are space parameters. For consistency define T1 as the time when the current reaches Lh, or something like that. . . 9) Figures 8, 9 and 10. Parameters L and Psi should have units.

References: Chikita, K., N. Yonemitsu, and M. Yoshida (1991), Dynamic sedimentation processes in a glacier-fed lake, Peyto Lake, Alberta, Canada, Jpn. J. Limnol., 52(1), 27–43, doi:10.3739/rikusui.52.27. Sedimentation engineering : processes, measurements, modeling, and practice. Marcelo H García Published in 2008 in Reston, Va. by American Society of Civil Engineers.

---

## Referee Comment (RC2) · O. Sequeiros (Referee) · 6 Mar 2018

I have reviewed the updated document by Zordan et al. The authors have have addressed my comments. I recommend this manuscript for publication. Octavio E. Sequeiros

---

## Author Comment (AC1) · 6 Mar 2018

Associate Editor We revised the manuscript to account for all the comments pointed out by the reviewer. In the following, we provide the answer to the specific points. Thanks to the precise and constructive comments, we hope the general quality of the manuscript, as well as the clarity of text and illustrations, has been improved.

The authors noticed that a unit typo had produced wrong values of slopes for Table 1, which are now corrected. Some text was modified to accommodate the changes which do not influence the discussion and final conclusions.

[Figure]

  Answer to O. Sequeiros The reviewer's suggestions are kindly acknowledged since they have an encouraging purpose of improving the manuscript. Reply to the queries made by the reviewer and the answer given by the authors are listed below. The English of the manuscript has been checked by a professional service.

General comments: 1) The word geomorphic in the title refers to change in morphology, including process of sediment entrainment, transport and deposition. It's probably too generic to be used in the title of the paper that mainly focus on the erosion process. The title was changed into: "Potential erosion capacity of gravity currents created by changing initial conditions". We have changed also in the abstract, line 7, "erosion potential" instead of "geomorphic". 2) We have specified in the introduction how the set-up is by adding the sentence: "The bottom of the channel was designed in order to have a variable slope angle of the lock and a following flat surface." A new reference has been added in the introduction: "Mulder, T. and Alexander, J.: Abrupt change in slope causes variation in the deposit thickness of concentrated particle-driven density currents, Marine Geology, 175, 221–235, 2001." In the discussion, modifications of the text are done as well in order to avoid misleading interpretations of the set-up configuration. 3) The function H(t)=ud(t)h(t) has been defined by considering both the gravity current contour and the depth averaged streamwise velocity because this represent a flow rate per unit width. The head is generally elevated with respect to the following body and it is also characterized by a core of intense streamwise velocity. Therefore, the function here defined takes into account both features. Finally, Nogueira et al. (2014) used a similar procedure which considered the depth averaged density instead of the velocity, hence defining the currents regions recurring to a measure of mass flux. The following sentence has been added in Section 3.1: "We can moreover notice that, by dimensional analysis, the function H corresponds to a flow rate per unit width." 4) The filtering process is actually filtering above the 8Hz frequency, this means that the filter passes (without modifications) signals with a frequency lower than the 8Hz. The following sentence has been added for clarity in Section 2.2: "The 8 Hz cut-off has been chosen because the signal, for frequencies higher than 8 Hz, showed

white noise." 5) The comparison is done looking at tests with lock-slope ($S_i$ tests) and correspondent tests with same lock-volume but on horizontal bed ($L_i$ tests). Figure 1 has been modified in order to clarify the parallel between the two sets-up: lock-slope and volume reduction on horizontal bed. The fact that results for tests $S_i$ show a reduction in the streamwise velocity is therefore not caused by the reduced volume of release but a consequence of the developing descending flow in the upstream reach of the channel. To precise this point, the sentence in Section 3.2 has been modified to "By comparing tests $S_i$ with the correspondent $L_i$ tests, which have the same lock-volume but are performed without upstream slope, it is noticed that mean streamwise velocity is slightly higher for tests on horizontal bed." 6) The reviewer is right and the sentence has been changed in the text. 7) The assumption of hydraulically smooth flow was indeed verified. The shear Reynolds number (or skin roughness, ks, normalized by the viscous layer) has be verified to be lesser than 5: ks u*/ $\nu$ <= 5 (Zordan et al.,2016). This is now in the manuscript. 8) In Zordan et al.,2016 the logarithmic profile method applied for one of the velocity profile is shown. The collapsed near-bed profiles of all tests following a line with equation u/u*=1/k(ln z/z0) is also shown in Zordan et al.,2016. This reference has therefore been added to the paper and a more complete explanation is made now. 9) We reformulated the misleading sentence in "Tests performed with a slope break at the section of the lock show..." 10) The computation of interface shear stress as proposed by Chikita et al (1991) requires estimation of the depth-averaged density which was not measured in the present study. The application of the regression curves for the estimation of the drag coefficients are derived for turbidity currents observed in a reservoir by Chikita et al (1991) whose application at our case study is arguable: these are physical factors that depends on the specificities of the case. 11) We took out the word "steady". In the configuration with the greatest volume of dense fluid released, quasi-steady conditions can be reached in the body region but, as pointed out by the reviewer, with shorter locks the flow is not steady but inherently transient. 12) and 13) The main reason for potential bottom erosion reduction in essentially the reduction in the volume of release. To make clearer this point

I've added the sentence in Section 4.3: "This is mainly the result of the released volume reduction caused by the presence of the lock-slope, therefore originating shorter current bodies". The reviewer is right in his comment but we realize that this is the product of misleading text. In fact, our objective was to verify how the process occurring in the current initiated still at the lock, which may be caused by different densities and with different inclinations, would influence the propagating current in a downstream horizontal reach. The introduction and the conclusions were edited to make this clear. The physical interpretation is kept but now adequately spatially contextualized. 14) We have rewritten the sentence in Section 5 as "Bottom erosion capacity is reduced by the presence of the extra gravitational forces, most probably due to lower streamwise velocities which are consequence of gravity currents dilution." 15) We took into account your comment and therefore the conclusions were introduced with a new paragraph (Section 5): "In most practical situations gravity currents are flowing on different topographies and most of the time travels along inclined but discontinuous slopes (slope breaks). Moreover they are generally originated by the release of a certain amount of a fluid of various densities. The present study these both changing initial conditions which trigger gravity currents that are commonly observed in nature.".

Minor/format comments: 1) The new sentence is: "The shape of the current is modified due to the enhanced entrainment of ambient water and the body is the region of the current where this most happens." 2) The new sentence is: "The implications of an inclined lock on the potential entrainment capacity of the flow is here discussed." 3) Typing error: "in some cases". 4) I've corrected the typing error in the bibliography: Niño. 5) Caption Table 1 has been corrected. 6) The definition of R-square is added. 7) "faster material" was changed to "faster fluid". 8) Lh and Lb are also time parameter and therefore the statement T1=Lh and T2 =Lb are dimensionally correct. Anyway we wanted to highlight the duality space-time, which is important to translate the herein temporal measurements into spatial measurements, as others works may use. Therefore, a sentence was added in line 5 chapter 3.1:" L_h identifies the temporal extension of the head. The conversion from time to length scale may be done

by using Taylor frozen hypothesis and considering a reference velocity of the current velocity as advection velocity." 9) Units have been added to the plots.

References: Chikita, K., N. Yonemitsu, and M. Yoshida (1991), Dynamic sedimentation processes in a glacier-fed lake, Peyto Lake, Alberta, Canada, Jpn. J. Limnol., 52(1), 27–43, doi:10.3739/rikusui.52.27. Mulder, T. and Alexander, J. (2001). Abrupt change in slope causes variation in the deposit thickness of concentrated particle-driven density currents, Marine Geology, 175, 221–235. Nogueira, H. I., Adduce, C., Alves, E., and Franca, M. J. (2014). Dynamics of the head of gravity currents. Environmental Fluid Mechanics, 14(2):519–540. Zordan J., Schleiss A.J. and Franca M.J. (2016). Bed shear stress estimation for gravity currents performed in laboratory. Proc. of River Flow 2016, St. Louis, USA, 855-861.

The revised manuscript is hereafter:

Please also note the supplement to this comment:
https://www.earth-surf-dynam-discuss.net/esurf-2017-63/esurf-2017-63-AC1-supplement.pdf

**Supplement:**

[revised manuscript text omitted]

---

## Referee Comment (RC3) · J. Eggenhuisen (Referee) · 23 Mar 2018

Many lock-box exchange experiments with gravity currents are performed with horizontal flume floors. While gravity currents over horizontal surfaces do exist, many gravity currents flow down a slope. In such cases the slope-parallel component of the gravitational force acting on the excess density acts to propel the flow down the slope, and a steady flow may be generated when the gravitational forces are balanced by friction with the bed and the ambient fluid. Many lock-box studies do not address the question whether the two hydrodynamic structures on flat beds and sloping beds are fundamentally different. This manuscript explicitly studies the consequences of the differences in hydrodynamic structures between gravity currents on horizontal beds and gravity currents flowing down a slope. It thereby addresses a question that I have long won-

dered about when reading papers about lock-box exchange experiments. I applaud this effort, and think it is a significant contribution to the literature.

However, I think considerable improvements are needed in the presentation of the results and clarification of the text.

The results look like intricate quantitative analyses of the measurements, but are presented in a way that makes it very difficult to assess the answer to the research question. The main results from figures 4-7 are difficult to read because they are so overwhelming with dominating fluctuations. These results are in need of syntheses and parameterisation, such that the reader can interpret the effect of slope on the flow structure. I would suggest to present a selection of key runs in full page width figures; and to leave out the detailed measurements for the other runs. Figure 8 is a good start, but I would suggest to first plot the shape variables (Lb,S) against the slope, such that the effect of slope on flow size can be assessed (and similar plots for Hb and the time averaged velocity and bed and interfacial shear stress).

My second main concern about the manuscript relates to the reliance on Zordan et al under review a, and Zordan et al. under review b, which are not available at this stage. The postulation of new entrainment variables in sections 4.2 and 4.3 does not seem justified without publication of those papers. Furhtermore, I find it problematic to discuss bottom erosion by flows in section 4.3, based on experiments that did not include erodible beds. This seems to be a step back from the seminal gravity current erosion experiments of Garcia and Parker (Garcia & Parker, 1991, 1993). Why is the entrainment relation arising from their work (or any other entrainment rate, e.g. Dorrell et al. 2018, GRL) not sufficient? Likewise, for the water entrainment, the new parameter in section 4.2 is not explained in sufficient detail for me to evaluate its merit. Why not use the available Ri-dependent criteria? I suggest to cut sections 4.2 and 4.3 from the manuscript.

Below I have formulated my further comments to the manuscript. I hope the authors

**ESurfD**
find these useful and constructive.

Yours Sincerely,

Joris Eggenhuisen.

Main Comments: - The postulation of new entrainment variables in sections 4.2 and 4.3 does not seem justified without publication of Zordan et al under review a, and Zordan et al. under review b, which are not available at this stage. Furhtermore, I find it problematic to discuss bottom erosion by flows in section 4.3, based on experiments that did not include erodible beds. This seems to be a step back from the seminal erosion experiments of Garcia and Parker (Garcia & Parker, 1991, 1993). Why is the entrainment relation arising from their work (or any other entrainment rate, e.g. Dorrell et al. 2018, GRL) not sufficient? Likewise, for the water entrainment, the new parameter in section 4.2 is not explained in sufficient detail for me to evaluate its merit. Why not use the available Ri-dependent criteria? I suggest to cut sections 4.2 and 4.3 from the manuscript.

-The structure of many sentences could be improved to increase the readability of the English text. I am clearly not a fluent native English writer myself, so I may not be the right person to point to style mistakes. An example of a sentence that I might suggest to be recast is the third sentence of the Introduction, which improves if the subject of the sentence is brought to the front: "Katabatic winds or sea breezes are examples of gravity currents in the atmosphere, in which the density gradients are caused by temperature inhomogeneities." Let me be clear to state that this comment does not reflect on the quality of the science presented in the paper.

-P2L19-24 I am not sure I understand the authors' perspective on the two flow dynamics states. In my understanding there are flows on horizontal planes that can be called momentum driven, these will always dissipate under action of friction as the flow spreads over the flat floor and finds a hydrostatic equilibrium with density gradients only in the vertical orientation. And there are flows down a gravitational gradient. If these

are steady, this implies a balance between gravitational driving force and friction on the flow, which is divided over the top and bottom boundaries (see for instance the definition sketch of the force balance in Konsoer et al (Konsoer, Zinger, & Parker, 2013). The dichotomy between friction-governed and gravitationally-governed is confusing to me.

- The frequency data in Figure 2 is not relevant to the results. It could be in the methods section, but I find the filtering and decomposition is already dealt with sufficiently there.

-P7 Figure 3. The variable H [mˆ2/s] is confusing me, please explain in more detail what it means, and how it is relevant to the research question of the manuscript. P8-12 Figs 4-7. The main results from figures 4-7 are difficult to read because they are so overwhelming with dominating fluctuations. These results are in need of syntheses and parameterisation, such that the reader can interpret the effect of slope on the flow structure. Figure 8 is a good start, but I would suggest to first plot the shape variables (Lb,S) against the slope, such that the effect of slope on flow size can be assessed (and similar plots for Hb and the time averaged velocity and bed and interfacial shear stress).

-P10 Fitting the logarithmic law of the wall to very thin gravity current boundary layers is notoriously difficult. Please report the precise approach taken. If I understand the workflow correctly z0 was the second free parameter in a two-parameter linear regression (the other being u*). Please report z0 and confirm that it had the correct size for hydraulically smooth flow.

-P11 Figure 6. The data in this figure is very difficult to read. Would it be possible to determine time-averaged shear stress values, and plot these against slope? Such a synthesis might be clearer than an accumulated set of panels showing all of the time series.

-P12L6 The definition of a novel ambient water entrainment variable is not clearly justified, and relies on the companion paper Zordan et al. under review a, which is not available to the readers of Earth Surface Dynamics at this time.
Minor comments: -The title is not clear; are the gravity currents created by changing conditions, or is the topic of the paper the geomorphic implication of changing initial conditions.

-P2L3 Traer et al. (Traer, Hilley, Fildani, & McHargue, 2012)is also a suitable reference here.

-P2L14 The initial trigger is emphasised to be of particular importance, but the experiments do not address the trigger of the flow. I suggest to rephrase this emphasis to align the statement better with the experiments and analyses performed.

-P2L22 The authors describe the common experimental observation that the head of gravity currents on steep slopes is fed by by the steady current in the body; Azpiroz-Zabala et al. (Azpiroz-zabala et al., 2017) have recently argued that this is a small scale experimental artefact and that real world turbidity currents in submarine canyons have a different structure.

Textual: -P1L15 not "confer" -P2L8 "Niño and Garcia"? -P2L16 Not clear, rephrase.

References used in this review: Azpiroz-zabala, M., Cartigny, M. J. B., Talling, P. J., Parsons, D. R., Sumner, E. J., Clare, M. A., . . . Pope, E. L. (2017). Newly recognized turbidity current structure can explain prolonged flushing of submarine canyons. Science Advances, 3(October), e1700200. Garcia, M., & Parker, G. (1991). Entrainment of bed sediment into suspension. Journal of Hydraulic Engineering, 117, 414–435. Garcia, M., & Parker, G. (1993). Experiments on the entrainment of sediment into suspension by a dense bottom current. Journal of Geophysical Research, 98, 4793–4807. http://doi.org/10.1029/92JC02404 Konsoer, K., Zinger, J., & Parker, G. (2013). Bankfull hydraulic geometry of submarine channels created by turbidity currents : Relations between bankfull channel characteristics and formative fl ow discharge, 118, 216–228. http://doi.org/10.1029/2012JF002422 Traer, M. M., Hilley, G. E., Fildani, A., & McHargue, T. (2012). The sensitivity of turbidity currents to mass and momentum exchanges between these underflows and their surroundings. Journal of Geophysical Research:

Earth Surface, 117(F1), n/a-n/a. http://doi.org/10.1029/2011JF001990

**ESurfD**

---

## Author Comment (AC2) · 18 Jun 2018

**Associate Editor**

We revised the manuscript to account for all the comments pointed out by the reviewer. In the following, we provide the answer to the specific points. Thanks to the precise and constructive comments, we hope the general quality of the manuscript, as well as the clarity of text and illustrations, has been improved and that this new version proves to be suitable for publication.

**Answer to J. Eggenhuisen**

The reviewer's suggestions are kindly acknowledged since they have an encouraging purpose of improving the manuscript. Reply to the queries made by the reviewer and the answer given by the authors are listed below.

*The results look like intricate quantitative analyses of the measurements, but are presented in a way that makes it very difficult to assess the answer to the research question. The main results from figures 4-7 are difficult to read because they are so overwhelming with dominating fluctuations. These results are in need of syntheses and parameterisation, such that the reader can interpret the effect of slope on the flow structure. I would suggest to present a selection of key runs in full page width figures; and to leave out the detailed measurements for the other runs. Figure 8 is a good start, but I would suggest to first plot the shape variables (Lb,S) against the slope, such that the effect of slope on flow size can be assessed (and similar plots for Hb and the time averaged velocity and bed and interfacial shear stress).*

*My second main concern about the manuscript relates to the reliance on Zordan et al under review a, and Zordan et al. under review b, which are not available at this stage. The postulation of new entrainment variables in sections 4.2 and 4.3 does not seem justified without publication of those papers. Furthermore, I find it problematic to discuss bottom erosion by flows in section 4.3, based on experiments that did not include erodible beds. This seems to be a step back from the seminal gravity current erosion experiments of Garcia and Parker (Garcia & Parker, 1991, 1993). Why is the entrainment relation arising from their work (or any other entrainment rate, e.g. Dorrell et al. 2018, GRL) not sufficient? Likewise, for the water entrainment, the new parameter in section 4.2 is not explained in sufficient detail for me to evaluate its merit. Why not use the available Ri-dependent criteria? I suggest to cut sections 4.2 and 4.3 from the manuscript.*

*Main Comments:*

*-The postulation of new entrainment variables in sections 4.2 and 4.3 does not seem justified without publication of Zordan et al under review a, and Zordan et al. under review b, which are not available at this stage. Furthermore, I find it problematic to discuss bottom erosion by flows in section 4.3, based on experiments that did not include erodible beds. This seems to be a step back from the seminal erosion experiments of Garcia and Parker (Garcia & Parker, 1991, 1993). Why is the entrainment relation arising from their work (or any other entrainment rate, e.g. Dorrell et al. 2018, GRL) not sufficient? Likewise, for the water entrainment, the new parameter in section 4.2 is not explained in sufficient detail for me to evaluate its merit.*
*Why not use the available Ri-dependent criteria? I suggest to cut sections 4.2 and 4.3 from the manuscript.*

At this point the two mentioned publications have been published and the references are therefore updated in the paper. The two mentioned papers are:
- Zordan, J., Juez, C., Schleiss, A. J., and Franca, M. J.: Entrainment, transport and deposition of sediment by saline gravity currents, Advances in Water Resources, 2018a.
- Zordan, J., Schleiss, A. J., and Franca, M. J.: Structure of a dense release produced by varying initial conditions, Environmental Fluid Mechanics, 2018b.

Section 4.2 and 4.3 have been modified in order to better explain the formulation of the potential entrainment parameters. We acknowledge the constructive comments on this part and we think that these two sections, with the following improvements, enrich the paper and worth to be conserved.

In Section 4.2 the following paragraph has been added "Kelvin-Helmholtz instabilities have a major role in provoking water entrainment. They take the form of vortical movements generated due to velocity shear at the interface between the two fluids. Since shear stress is determinant in the process of water entrainment, a new quantity to account for the potential entrainment capacity of the gravity current is here defined on the base of the computed time evolution of the interfacial shear stress ($\tau\_m$)."

In Section 4.3 the following paragraph has been added "The magnitude of the shear stress at the lower boundary layer determines the sediment transport capacity of saline currents and whether erosion or deposition processes dominate the regime at the bottom boundary (Cossu and Wells, 2012).
Therefore, similarly to the interfacial water entrainment capacity, the bottom entrainment capacity, that can also be called the erosion capacity, is here defined on the base of the computed bed shear stress.
…
In Zordan et al. (2018a) this quantity has been estimated for gravity currents simulated in presence of an erodible bed. A relationship between the eroded volume of sediments provoked by the passage of the gravity current and $\Phi b$ has been found, therefore confirming that $\Phi b$ is a good estimator of the entrainment capacity of these flows."

***-The structure of many sentences could be improved to increase the readability of the English text. I am clearly not a fluent native English writer myself, so I may not be the right person to point to style mistakes. An example of a sentence that I might suggest to be recast is the third sentence of the Introduction, which improves if the subject of the sentence is brought to the front: "Katabatic winds or sea breezes are examples of gravity currents in the atmosphere, in which the density gradients are caused by temperature inhomogeneities." Let me be clear to state that this comment does not reflect on the quality of the science presented in the paper.***

The English of the paper has been reviewed. The sentence proposed has been, among others, improved and rewritten as "Temperature inhomogeneities cause the density gradient which is at the origin of katabatic winds, an example of the occurrence of gravity currents in the atmosphere.".

***-P2L19-24 I am not sure I understand the authors' perspective on the two flow dynamics states. In my understanding there are flows on horizontal planes that can be called momentum driven, these will always dissipate under action of friction as the flow spreads over the flat floor and finds a hydrostatic equilibrium with density gradients only in the vertical orientation. And there are flows down a gravitational gradient. If these are steady, this implies a balance between gravitational driving force and friction on the flow, which is divided over the top and bottom boundaries (see for instance the definition sketch of the force balance in Konsoer et al (Konsoer, Zinger, & Parker, 2013). The dichotomy between friction-governed and gravitationally-governed is confusing to me.***

The authors agree with the reviewer on this analysis but we should observed that the gravity currents reproduced in this paper are unsteady. Therefore friction and gravitational forces are not under an equilibrium condition. The currents originated down a slope gain momentum due to the incremental gravitational forces in the direction of the motion while, under a critical slope angle, a friction governed regime forms and tends to dissipate the current. The high unsteadiness of these flows is an indicator of the change in the governing forces which result from the changing initial conditions.

*- The frequency data in Figure 2 is not relevant to the results. It could be in the methods section, but I find the filtering and decomposition is already dealt with sufficiently there.*

Section 2.3 is part of the main Section 2 (Methodology). The author respectfully disagree and prefer to keep the Section. During the preparation of this paper the author encounter difficulties on the characterization of a mean velocity field for these highly unsteady gravity currents, a concern shared by other authors. We believe that a paper should present all crucial contents which ideally allow to reproduce the study and therefore the data filtering procedure worth to be explained.

*-P7 Figure 3. The variable H [m^2/s] is confusing me, please explain in more detail what it means, and how it is relevant to the research question of the manuscript. P8-12 Figs 4-7. The main results from figures 4-7 are difficult to read because they are so overwhelming with dominating fluctuations. These results are in need of syntheses and parameterization, such that the reader can interpret the effect of slope on the flow structure. Figure 8 is a good start, but I would suggest to first plot the shape variables (Lb,S) against the slope, such that the effect of slope on flow size can be assessed (and similar plots for Hb and the time averaged velocity and bed and interfacial shear stress).*

In Section 3.1 the following paragraph has been added in order to explain how function H has been derived "The head and the body can be distinguished by different velocity fields and shape. These distinctive features (a characteristic velocity and the contour of the current) have therefore been considered to identify the regions of the currents.
An interpretation of the function H has been added by the following paragraph: "By dimensional analysis, the function H corresponds to a flow rate per unit width. The head of the gravity current is characterized by a high specific flow rate which decreases at the rear of the head, a region where fluid is recirculated through vortical movements.

In order to clarify the need of identifying head and body regions the following paragraph has been added to Section 3.1: "Once those regions are univocally identified, the estimation of the influence of the slope on the variation of the shape and extension of head and body regions will therefore be possible."

*-P10 Fitting the logarithmic law of the wall to very thin gravity current boundary layers is notoriously difficult. Please report the precise approach taken. If I understand the workflow correctly z0 was the second free parameter in a two-parameter linear regression (the other being u*). Please report z0 and confirm that it had the correct size for hydraulically smooth flow.*

A new paragraph which detailed explain the application of the fitting procedure has been added to Section 3.3 as follow: "The equation of the logarithmic law of the wall can be rewritten as:

$$u = A\ln(z) - B$$

where

$$A = \frac{u_*}{\kappa}, \quad B = \frac{u_*}{\kappa}\ln(z_0)$$

Then, by determining the coefficients A and B through a fitting procedure, one obtains an estimation of $u_*$ which is the velocity scale corresponding to the bed shear stress."

The verification of the hydraulically smooth conditions have been made by checking the shear Reynolds number as reported in the paragraph: "The flow boundary is assumed to be smooth, as verified by the estimation of the shear Reynolds number (or skin roughness, ks, normalized by the viscous layer) is lesser than 5:

$$\frac{k_s u_*}{\nu} \leq 5 \,"$$

This hypothesis has been checked for each time step as the fitting procedure was implemented for all mean velocity profiles.

**-P11 Figure 6. The data in this figure is very difficult to read. Would it be possible to determine time-averaged shear stress values, and plot these against slope? Such a synthesis might be clearer than an accumulated set of panels showing all of the time series.**

The authors have followed the useful suggestion of the reviewer and added Figures 6 and 7 to the paper. Figure 8 and 9 have been reduced to show only one significant configuration for which the time evolution (in particular the behavior in the body region) is of interested to be report in details. The comparison between tests performed under different configurations have been done through the computation of the ratio between time-averaged shear stress of respectively tests with varying lock-lengths and test with varying lock-slopes ($\overline{\tau_L}/\overline{\tau_S}$). Section 3.3 has been deeply modified.

**-P12L6 The definition of a novel ambient water entrainment variable is not clearly justified, and relies on the companion paper Zordan et al. under review a, which is not available to the readers of Earth Surface Dynamics at this time.**

The paper which was at the date under review is now available and published as:

Zordan, J., Schleiss, A. J., and Franca, M. J.: Structure of a dense release produced by varying initial conditions, Environmental Fluid Mechanics, 2018b.

**Minor comments:**
**-The title is not clear; are the gravity currents created by changing conditions, or is the topic of the paper the geomorphic implication of changing initial conditions.**
The new title is "Potential erosion capacity of gravity currents created by changing initial conditions".

**-P2L3 Traer et al. (Traer, Hilley, Fildani, & McHargue, 2012) is also a suitable reference here.**
The reference has been added to the introduction within the following paragraph: "A proper parametrization of both upper layer and bottom entrainment is still an open research field which needs to be addressed. Indeed authors know that small variations in the entrainment parameters highly influence the flow dynamics (Traer et al.2012)".

**-P2L14 The initial trigger is emphasised to be of particular importance, but the experiments do not address the trigger of the flow. I suggest to rephrase this emphasis to align the statement better with the experiments and analyses performed.**

The sentence has been rephrased as: "To understand how lock-volume and lock-slope, which are initial trigger conditions of gravity currents, are linked with their transport capacity is thus of fundamental

importance." Therefore we highlighted that two specific initial conditions variations are studied (lock-volume and lock-slope).

*-P2L22 The authors describe the common experimental observation that the head of gravity currents on steep slopes is fed by the steady current in the body; Azpiroz-Zabala et al. (Azpiroz-zabala et al., 2017) have recently argued that this is a small scale experimental artefact and that real world turbidity currents in submarine canyons have a different structure.*

The authors thank for the notice of this interesting paper. The structure of the gravity currents we reproduced is indeed similar to the one recognized by many authors working on this field. Azpiroz-Zabala et al. have provided a new model for turbidity current structure that seems to detach from previous knowledge. In the flows they observed, the head, which they called "frontal-cell" is highly erosive being therefore able to self-sustaining itself and consequently to outrun the slower moving body of the flow. A stretching of the current occurs. This is the main difference between our results and the turbidity currents as described by Azpiroz-Zabala et al. The gravity currents here reproduce are conservative and produced by lock-exchange. There is not extra-buoyancy gained by incorporation of sediments or by the sustainment of a continuous release. Therefore this is, as explained in Azpiroz-Zabala et al., the main reason of the different shape developed by the currents.

This subject has been introduced in the Introduction and the following paragraph has been added: "Recently Azpiroz-Zabala et al. provided a new model for the gravity current structure. They argued that real world turbidity currents in submarine canyons are instead characterized by a so-called "frontal-cell" which is highly erosive and therefore able to self-sustaining itself and to outrun the slower moving body of the flow, creating a stretched current. Nevertheless, authors working on small scale experimental reproduces gravity currents agree on describing the shape of the gravity current as composed by an arising highly turbulent front, called head, followed by, in some cases, a body and a tail."

*Textual:*
*-P1L15 not "confer" –* replaced with "produces"

*P2L8 "Niño and Garcia"?* corrected

*-P2L16 Not clear, rephrase.* "Turbidity currents are of particular interest reservoirs sedimentation, which have important economic costs due to the loss of volume for water storage.."

*References used in this review:*

*Azpiroz-zabala, M., Cartigny, M. J. B., Talling, P. J., Parsons, D. R., Sumner, E. J., Clare, M. A., Pope, E. L. (2017). Newly recognized turbidity current structure can explain prolonged flushing of submarine canyons. ScienceAdvances, 3(October), e1700200.*

*Garcia, M., & Parker, G. (1991). Entrainment of bed sediment into suspension. Journal of Hydraulic Engineering, 117, 414–435.*

*Garcia, M., & Parker, G. (1993). Experiments on the entrainment of sediment into suspension by a dense bottom current. Journal of Geophysical Research, 98, 4793–4807.*
*http://doi.org/10.1029/92JC02404*

*Konsoer, K., Zinger, J., & Parker, G. (2013). Bankfull hydraulic geometry of submarine channels created by turbidity currents: Relations between bankfull channel characteristics and formative flow discharge, 118, 216–228.*
*http://doi.org/10.1029/2012JF002422*

[revised manuscript text omitted]

---

## Referee Report (RR1)

Review of 'Potential erosion capacity of gravity currents created by changing initial conditions' by Zordan et al. 2018.

**Summary**

This experimental paper employs a lock-exchange method to model saline gravity flows. Slope and flow density are varied to assess their influence on flow dynamics and morphology. The measurements and approaches taken to analyse the gravity flows are for the most part appropriate and sensible. The results appear to show a counterintuitive relationship between slope and flow speed/basal shear stress. This is attributed to a significant increase in upper surface entrainment with increasing slope.

However, currently there are some significant issues with the paper that need addressing before it is suitable for publication. These issues are outlined in detail below.

**Main points**

**Writing style**

I appreciate that English is not your first language, and am always impressed that people can write to such a high standard in their 2nd or 3rd language. Nevertheless, there are several non-sense sentences and several typos that need correcting. Please check again throughout the manuscript for these.

**Structure**

The aims of the paper are not clear. What is the bigger picture or generic learnings that the paper wants to address? Currently, it is difficult to pick out how the introduction links with the discussion and conclusions. This means that whilst there might be some good analysis made on your gravity flows, I don't see the point, i.e. how can the wider community use this work. This issue with structure feeds into several points below regarding methodology. For example, a major point in the paper is the analysis of entrainment dynamics, yet the introduction does not tell me why entrainment dynamics are important to understand and what key gaps in our knowledge there are.

**Methodology**

4.2 – ambient entrainment.

This section presents a revised method for the estimation of ambient water entrainment. However, there is no justification as to why this revised approach is needed. As the section states, there is a great deal of work that has established entrainment dynamics with Gradient Richardson numbers. It is not clear why you need to use a 'surrogate' formulation when you could just as easily use well-established methods. Make this justification explicit in this section.

**Results**

There is a counterintuitive relationship between slope and flow speed. This is due to the significant increase in ambient entrainment as the slope increases. I think this may well be a product of the experimental set up. In the description of the flume tank set up, you state that the flume is filled to a depth of 0.2 m. Does this mean that when you put a steeper slope into the flume the top parts of the ramp (i.e. up to where the lock box sits) are in shallower water? This would mean that the same lock-box volume is being released into the tank but the thickness of the overlying ambient water column is reduced, which would increase the velocity of the return flow. This increase in return flow velocity (due to the modification of the slope) will drive increased upper surface entrainment. This is not so much a product of the relationship between slope and entrainment but the influence of focussing a high(er) velocity return flow over the gravity current. This aspect of the experimental set up needs to be explained clearly and if this is the case, then the results require discussion in this light.

4.3 - Bottom erosion capacity

I do not understand this method of estimating erosion capacity. Bed shear stress is a measure of erosion in that it is related to the critical shear stress of particle movement (and/or flow capacity). This depends on the particle size, distribution and bed roughness parameters. You can talk about bed shear stresses changing in your experiments, and how this might influence erosion patterns but as written the discussion reads as though the flows passed over erodible beds. They don't.

Details

Typos

Figure 6,7 captions – 1[st] sentence '…respectively' doesn't make sense.

Chris Stevenson

---

## Author Response (AR2)

**Associate Editor**

We revised the manuscript to account for all the comments pointed out by the third reviewer. In the following, we provide the answer to the specific points. Thanks to the precise and constructive comments, we hope the general quality of the manuscript, as well as the clarity of text and illustrations, has been improved now making it suitable for publication.

**Answer to Chris Stevenson**

**The reviewer's suggestions are kindly acknowledged since they have an encouraging purpose of improving the manuscript. Reply to the queries made by the reviewer and the answer given by the authors are listed below.**

1. **Writing style**

The English of the paper has been reviewed.

2. **Structure**

*The aims of the paper are not clear. What is the bigger picture or generic learnings that the paper wants to address? Currently, it is difficult to pick out how the introduction links with the discussion and conclusions. This means that whilst there might be some good analysis made on your gravity flows, I don't see the point, i.e. how can the wider community use this work. This issue with structure feeds into several points below regarding methodology. For example, a major point in the paper is the analysis of entrainment dynamics, yet the introduction does not tell me why entrainment dynamics are important to understand and what key gaps in our knowledge there are.*

The importance of the entrainment dynamics has been explained by the following sentences, which have been added to the paper: "Small variations in the entrainment highly influence the flow dynamics (Traer et al., 2012). Due to the instabilities at the interface with the ambient fluid, the current entrains the lighter fluid and therefore it dilutes. "

The aims of the paper have been emphasized and made more explicit in the text particularly in the Introduction. The following sentences have been added: "To understand how lock-volume and lock-slope, which are initial trigger conditions of gravity currents, are linked with their transport capacity is thus of fundamental importance and it is the main objective of this paper. We show how shear stress at the boundaries is dependent to the set-up under which a gravity current forms, i.e. its initial and boundary conditions. Different initial conditions, representing configurations which can possibly be found in nature, are tested by varying the initial volume of denser fluid, and the lock geometry."

The gaps of knowledge that this paper aims to fill have been explicitly mentioned through the following sentences: "We were in search for a threshold at which an inversion of the leading forces of these currents would occur, which are gravitational forces and friction at the upper interface with the ambient fluid. Previous studies mainly focused separately on either low slopes or large slopes, missing the analysis of the transition which is here tested thanks to a specific experimental set-up which allows a wider range of configurations. Finally, we use a parameter previously defined in Zordan et al. (2018) for the evaluation of the bottom erosion capacity, as a surrogate to evaluate the influence of each different trigger condition on the erosion capacity of the currents."

3. **Ambient entrainment**

   *This section presents a revised method for the estimation of ambient water entrainment. However, there is no justification as to why this revised approach is needed. As the section states, there is a great deal of work that has established entrainment dynamics with Gradient Richardson numbers. It is not clear why you need to use a 'surrogate' formulation when you could just as easily use well-established methods. Make this justification explicit in this section.*

The advantages of using this model for water entrainment is the subject of the cited paper (Zordan et al., 2018b). The following sentence has been added in order to explain the main differences between the models: "Since bulk Richardson number is based on depth averaged quantities, it assumes that properties do not vary significantly along the vertical.

The quantity $\phi_m$, a surrogate for entrainment capacity, relies to the instantaneous measurements of shear stress and therefore account for the unsteady behaviour of the currents."

4. **Results**

   *There is a counterintuitive relationship between slope and flow speed. This is due to the significant increase in ambient entrainment as the slope increases. I think this may well be a product of the experimental set up. In the description of the flume tank set up, you state that the flume is filled to a depth of 0.2 m. Does this mean that when you put a steeper slope into the flume the top parts of the ramp (i.e. up to where the lock box sits) are in shallower water? This would mean that the same lock-box volume is being released into the tank but the thickness of the overlying ambient water column is reduced, which would increase the velocity of the return flow. This increase in return flow velocity (due to the modification of the slope) will drive increased upper surface entrainment. This is not so much a product of the relationship between slope and entrainment but the influence of focussing a high(er) velocity return flow over the gravity current. This aspect of the experimental set up needs to be explained clearly and if this is the case, then the results require discussion in this light.*

The gravity current forms the instant the gate is opened. The head shapes at the slope-break and the current develops along the horizontal bottom where the thickness of the clear water is always the same for all tests. To clarify this point the following sentence has been added to Chapter 4.2: "The enhanced entrainment which has been verified for gravity currents formed downstream steep slopes is due to the shear at the interface with the ambient water. Gravity currents are likely experiencing two phases while flowing along the channel. An initial acceleration takes place due to the higher gravitational forces, then the current accelerates inducing an increment of shear stresses at the interface. The entrainment of clear water is therefore intensified and the current are diluted. At the point where the measurements are taken, the gravity currents are experiencing this second phase."

5. **Bottom erosion capacity**

*I do not understand this method of estimating erosion capacity. Bed shear stress is a measure of erosion in that it is related to the critical shear stress of particle movement (and/or flow capacity). This depends on the particle size, distribution and bed roughness parameters. You can talk about bed shear stresses changing in your experiments, and how this might influence erosion patterns but as written the discussion reads as though the flows passed over erodible beds. They don't.*

Indeed, the experiments presented in this manuscript are performed over a fixed bed. We use however the parameter defined in Zordan et al. (2018), where experimental work to characterize sediment entrainment by the passage of a saline current over erodible bed is analysed, as a surrogate to see here the effects of the changes in the initial lock geometry and current initial buoyancy. We added information on this replying to #1 and also further in the text as follows:

"Although the present experiments are over a fixed bed, this estimator will be used here to evaluate the influence of the lock initial conditions in the entrainment capacity of these flows."

**6. Typos**

Typos have been corrected.

---

## Author Response (AR3)

**Associate Editor**

We appreciate the comments to our paper and we hope to answer through this document the last points which have been identified as needing more clarity.

Our answers to the specific comments on the response to Reviewer 2 are as following:

*4. "Gravity currents are LIKELY experiencing two phases...an initial acceleraton takes place...then the current accelerates...At the point where the measurements are taken, the gravity currents are experiencing this second phase."*

*-> It seems that this is something that the authors should know, rather than guess about. I get that the ADV measurements are at a fixed point, but the aren't there at least movies or photographs showing the propagation of the head in order to deduce the front speed through time?*

Unfortunately we did not record any video of the tests which could help deducing the front speed. Since this information is missing, the discussion concerning the initial acceleration is a hypothesis supported by previous experiments as discussed by Beghin et al. (1981). We changed the sentence to:

"According to Beghin et al. (1981), gravity currents are experiencing two phases while flowing along the channel."

*-> As for the reviwer's concern about return flow, this could be checked with the velocity profile, right? In other words, if flow velocity becomes negative at or above the current interface - and if this effect becomes larger for steeper slopes - then the reviewer's point is valid. Showing that this does NOT happen would demonstrate that the reviewer's concern is not important. From figure 5 it looks like there is significant return flow, and that it might be dependent on slope. Please explain.*

The return flow cannot be influenced by the set-up since it has been conceived in order to have the same water depth for all tests (0.2m). The velocity field shown in Figure 5 is anyway confirming that the mean velocity of the return flow for tests performed with the inclined bottom is higher than for the correspondent tests on the horizontal. As discussed in the paper, the gravity currents flowing an inclined bottom experience subsequent phases: (i) at the beginning an acceleration as a consequence of their increased gravitational forces; (ii) the acceleration induces the acceleration of the return flow and it enhances ambient water entrainment from the interface which causes the dilution and, accordingly, (iii) a deceleration of the gravity current. Higher return flows velocities prove the passage through phase (ii).

*General comment on "readability"*

The paper have been reviewed and many paragraph have been improved in order to improve their understanding.

*Specific comments*

*Abstract:*

*Some motivation sentence should come FIRST, before the sentence that states the experiments that were run.*

*Experimental sentence is not clear, for example: "...for horizontal and four inclinations".*

*The second sentence (motivation) is also unclear: you talk about changing the slope upstream of a lock gate, but the reader has not been introduced to the experiment so does not know what this means. It seems to me that the motivation is that you want to examine the influence of initial conditions of a gravity current on its runout and entrainment capacity over a horizontal bed. You can then explain how you modify these initial conditions; in this experiment you examine primarily the slope. ALSO, your sentence discussing the main point of the experiments does not agree completely with the title; for consistency, you should talk about transport capacity or entrainment potential, but NOT BOTH. It becomes confusing to the reader.*

*The rest of the abstract does not include results and a conclusion; it talks about what will be discussed in the paper, rather than summarizing the paper. As such, the abstract is not that helpful. I suggest a re-structuring:*

*1. Motivation of the runout and entrainment of gravity currents, and how the initial conditions may govern these dynamics - including pointing out something that is uknown or unexpolored.*

*2. Then say that you have built an experiment to explicitly test the control of some factors (identified in the first sentence) on the flow dynamics and sediment entrainment potential of gravity currents. You should also make clear that your gravity currents are saline and nor particulate.*

*3. What was the range of conditions explored?*

*4. What were the main findings?*

*5. End with a sentence about the implications of your work for broader considerations of turbidity currents.*

The order of the sentences, the grammar and the incongruences of terminology have been corrected by a complete revision of the Abstract which consider these last 5 points.

"We investigate to which extent initial conditions (in terms of buoyancy and geometry) of saline gravity currents flowing over a horizontal bottom influence their run-out and entrainment capacity. In particular, to which extent the effect of the introduction of an inclined channel reach, just upstream from the lock gate, influences the hydrodynamics of gravity currents and consequently its potential erosion capacity is still an open question. The investigation herein presented focus on the unknown effects of an inclined lock on the geometry of the current, on the streamwise velocity, on bed shear stress and on the mechanisms of entrainment and mass exchange. Gravity currents were reproduced in laboratory, through the lock-exchange technique, and systematic tests were performed with different initial densities, combined with five initial volumes of release on horizontal and sloped locks. The inclination of the upstream reach of the channel (the lock) was varied from 0% to 16% while the lock-length was reduced up to ¼ of the initial

reference case. We observed that the shape of the current is modified due to the enhanced entrainment of ambient water, being the body the region of the current where this most happens. A counter-intuitive relation between slope and mean streamwise velocity was found, supporting previous findings which hypothesized that gravity currents flowing down small slopes experience an initial acceleration followed by a deceleration. For the steepest slope tested, two opposite mechanisms of mas exchange are identified and discussed, i.e. the current entrainment of water from the upper surface due to the enhanced friction at the interface and the head feeding by a rear fed current. The bed shear stress and the corresponding potential erosion capacity are discussed giving insights into the geomorphological implications of natural gravity currents caused in different topographies settings."
* * *
**Introduction**

Multiple sentences have been modified all along the paper and particularly in the introduction, resulting in a considerable enhancement of the manuscript readability.

***-First sentence is almost impossible to read.***

The sentence have been rewritten: "Gravity currents are common phenomena which may occur spontaneously in nature or triggered by human activities."

***-Many of these sentences are written in a rather indirect way. Sentence structure should be more simple, direct and make use of more straightforward words. For example, I notice the authors changed the sentence on katabatic winds from the first draft to make it more readable, but it is now even LESS readable. A better style would be "Katabatic winds are an example of an atmospheric gravity current, which arises due to...".***

The sentence has been rewritten: "Examples of gravity currents generated in the atmosphere are katabatic winds which are created by temperature inhomogeneities that originate the density gradient.".

***-Sand storms are NOT examples of particulate gravity currents. The sand concentrations are too low to contribute to density contrast. Rather, cold atmospheric currents can be gravity currents that pick up sand and dust - but the particles act as tracers of the flow, not drivers of it.***

The example of sand storms have been removed.

***-Top of page 2: "...able to self-sustaining itself and to outrun the slower moving body of the flow..."***

***First, "self-sustaining itself" must be changed. Second, how does this new idea relate to the classic Parker work of auto-suspension and also the process they have outlined for heads breaking away from the bodies of currents?***

Azpiroz-Zabala et al. 2017 defined a new turbidity current structure which substantially differs from what has been presented by Parker. Azpiroz-Zabala et al. said that in many laboratory-scale flows, as it has been performed by Parker, "short-lived experimental flows (termed surges)" generates in which "the head does not outrun the body, and the body is rather poorly developed".

The following sentence have been added to the introduction for clarifying the two positions: "The difference between the two concepts mainly comes from the observation time frame which is of the order

of hours for the experimentally reproduced gravity currents while it's of days for the observations that Azpiroz-Zabala et al. (2017) made in Congo Canyon.".

*p.2 line 18: The lock-exchange component is an experimental initial condition. The paper makes it sound as if lock conditions are an important component for gravity currents in nature. They are not. Rather, experiments choose a lock exchange geometry simply because it is controllable and reproducible. So it is not of "fundamental importance" generally for gravity currents.*

The sentence has been modified: "Lock-volume and lock-slope are initial trigger conditions of the experimentally reproduced gravity currents and the main objective of the paper is to understand their influence into the transport capacity of the flows.".

*p. 2 bottom: I do not know what "increment of friction means".*

The expression "increment of friction" has been replaced with "enhanced friction at the interface" or "enhanced shear stress".

*p.4. Please define all variables as they are presented.*

A list of symbols with their definition and unit of measures have been added to the paper.